# FCM: FORGETFUL CAUSAL MASKING MAKES CAUSAL LANGUAGE MODELS BETTER ZERO-SHOT LEARNERS

## ABSTRACT

Large language models (LLM) trained using the next-token-prediction objective, such as GPT3 and PaLM, have revolutionized natural language processing in recent years by showing impressive zero-shot and few-shot capabilities across a wide range of tasks. In this work, we propose a simple technique that significantly boosts the performance of LLMs without adding computational cost. Our key observation is that, by performing the next token prediction task with randomly selected past tokens masked out, we can improve the quality of the learned representations for downstream language understanding tasks. We hypothesize that randomly masking past tokens prevents over-attending to recent tokens and encourages attention to tokens in the distant past. By randomly masking input tokens in the PaLM model, we show that we can significantly improve 1B and 8B PaLM's zero-shot performance on the SuperGLUE benchmark from 55.7 to 59.2 and from 61.6 to 64.0, respectively. Our largest 8B model matches the score of PaLM with an average score of 64, despite the fact that PaLM is trained on a much larger dataset (780B tokens) of high-quality conversation and webpage data, while ours is trained on the smaller C4 dataset (180B tokens). Experimental results show that our method also improves PaLM's zero and few-shot performance on a diverse suite of tasks, including commonsense reasoning, natural language inference and cloze completion. Moreover, we show that our technique also helps representation learning, significantly improving PaLM's finetuning results.

## 1 INTRODUCTION

Language model (LM) pre-training has substantially advanced the state-of-the-art across a variety of natural language processing tasks (Peters et al., 2018; Devlin et al., 2018; Brown et al., 2020; Chowdhery et al., 2022) and related fields including image generation, reasoning, and code generation (Alayrac et al., 2022; Lewkowycz et al., 2022; Saharia et al., 2022; Chen et al., 2021). Prior work on pre-training have focused on mixing different choices of architecture (*e.g.*, encoder-only, decoder-only, or encoder-decoder) with different objective functions (*e.g.*, masking or causal language modeling). For example, masked encoder-only models such as BERT (Devlin et al., 2018) and RoBERTa (Liu et al., 2019) excel in discriminative finetuning tasks such as classification. Similarly, masked encoder-decoder models such as BART (Lewis et al., 2019) and T5 (Roberts et al., 2019) perform well on both discriminative and generative finetuning. While masked language modeling is effective for finetuning and removes the need for task-specific architectures, its major limitation is that there is still a need for task-specific datasets and task-specific finetuning. On the other hand, decoder-only causal language models remove such limitations. In fact, they are capable of zero-shot and few-shot adaptation without the need for finetuning, by simply prompting the model with appropriate strings to control the generated outputs, as shown in GPT3 (Brown et al., 2020) and PaLM (Chowdhery et al., 2022).

Driven by its impressive zero-shot and few-shot abilities, there has been more work on scaling causal decoder-only architectures (Zhang et al., 2022; Black et al., acl; Brown et al., 2020; Chowdhery et al., 2022) compared to encoder-based architectures, and there has been significant interests in studying such models in various contexts (Hoffmann et al., 2022; Wei et al., 2022b; Li & Liang, 2021; Ahn et al., 2022; Chen et al., 2021). However, such decoder-only models are still limited by their imperfect zero-shot and few-shot adaptation compared to human performance, and their relatively inferior finetuning performance compared to masked language modeling.

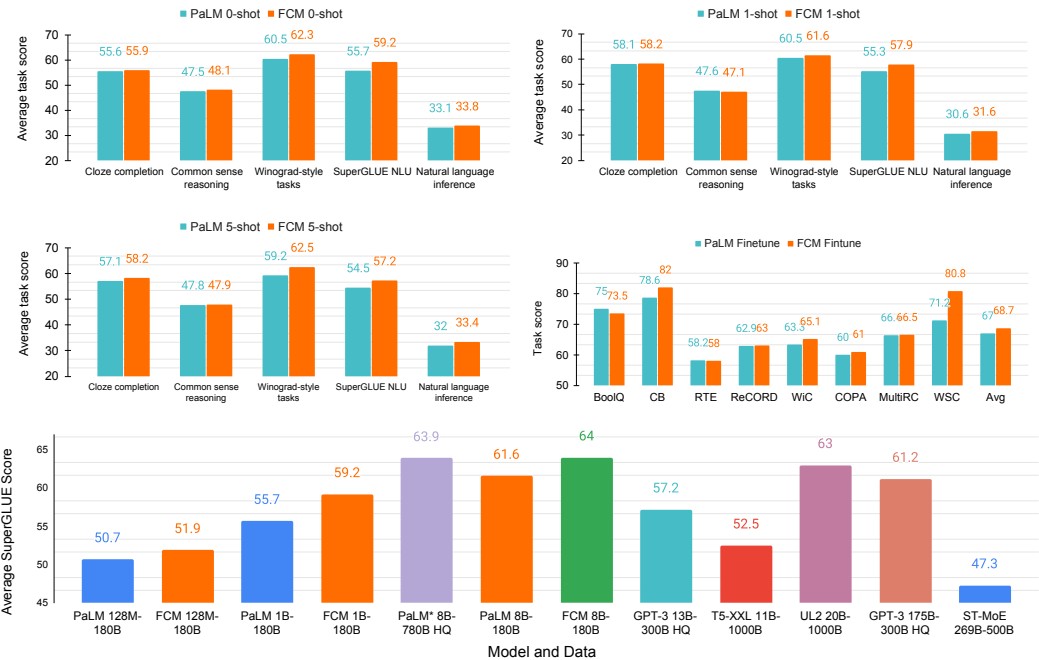

Figure 1: FCM outperforms PaLM in zero- and few-shot as well as finetuning tasks. **Top & middle**. Task average performance grouped by categories. The model size is 1B. We report the averaged scores in each category. Scores are averaged over 3 evaluation random seeds. **Bottom**. SuperGLUE zero-shot performance by different model size and dataset size. PaLM⋆ 8B-780B HQ denotes the published results of 8B model trained on 780B tokens from high quality datasets, PaLM 8B-180B denotes the same setup but with 180B tokens from C4 dataset, and FCM 8B-180B denote the same 8B model trained on 180B tokens from C4 dataset using FCM as objective.

To address the above challenges, prior work have proposed to combine masked modeling with causal language modeling (Dong et al., 2019; Wang et al., 2022; Tay et al., 2022; Du et al., 2022) to bring the benefit of masked modeling to causal language models while retaining their zero-shot ability. However, such approaches typically introduce extra computation and parameters or require using a sophisticated attention masking strategy which hinders practical usages (Yang et al., 2019; Tay et al., 2022). Moreover, they typically train encoder-decoder models which are not naturally suitable for zero- and few-shot inference tasks compared with decoder-only causal language models and are still outperformed by causal language models (Sanh et al., 2022; Brown et al., 2020; Chowdhery et al., 2022). In order to further improve causal language models few-shot abilities, some works proposed better prompt engineering methods (Liu et al., 2021; Lester et al., 2021; Ling et al., 2017; Wei et al., 2022b; Li & Liang, 2021) or better finetuning methods (Mishra et al., 2022; Wei et al., 2022a; Sanh et al., 2022). Prompt-based methods are sensitive to design (Lester et al., 2021; Liu et al., 2021), while finetuning-based approaches typically require a huge amount of supervision to work with as shown in Sanh et al. (2022). In addition, such methods can only improve pre-trained model and are unable to improve pre-training.

In this work, we propose a pre-training approach that does not incur any extra computation cost or parameters, to improve few-shot and zero-shot performance, as well as representation learning of causal language models. Our key observation is that, by performing next token prediction task with randomly selected past tokens masked out, we can improve the quality of the learned representations for downstream language understanding tasks. Our method, Forgetful Causal Masking (FCM), can be efficiently implemented by randomly masking input tokens in the causal language model. Applying our method to PaLM (Chowdhery et al., 2022), a state-of-the-art causal language model, we see significant improvement on the SuperGLUE (Sarlin et al., 2020) benchmark: our method significantly improves the 1B-model-size PaLM's zero-shot performance from 55.7 to 59.2 and improves the 8B-model-size PaLM's zero-shot performance from 61.6 to 64.0. We also conduct extensive evaluation on the commonsense reasoning benchmark PIQA (Bisk et al., 2019), ARC (Yadav et al.,

2019), and OpenBookQA (Mihaylov et al., 2018); the Winograd-style tasks Winograd (Sakaguchi et al., 2020) and WinoGrande (Kocijan et al., 2020); the natural language inference (NLI) benchmark ANLI (Nie et al., 2019); and cloze completion tasks StoryCloze (Mostafazadeh et al., 2016) and LAMBADA (Paperno et al., 2016); and find that our method improves the zero-shot and few-shot performance of PaLM on all of the diverse suite of tasks. In addition, FCM improves representation learning, as shown in our SuperGLUE finetuning experimental results, where our method significantly improves 1B parameter PaLM model's finetuneing performance from 67.0 to 68.7, and our method significantly improves 8B parameters PaLM model's finetuning performance on all 8 SuperGLUE tasks, improving the score from 80.7 to 83.1.

**Contributions.** We highlight the contributions of our paper below:

- We present FCM, a simple and scalable pre-training methodology for causal language modeling. We provide the empirical evaluation of FCM on a suite of few-shot and finetuning benchmarks.

- We show that FCM is highly effective at improving zero-shot and few-shot learning results, outperforms strong baselines including PaLM and UL2, improving the average SuperGLUE score of 8 billion parameters PaLM from 61.6 to 64.0, and improving PaLM on a wide range of 19 NLP tasks.

- In addition to few-shot learning, we demonstrate that FCM significantly helps with finetuning to downstream tasks, improving the performance of 8 billion parameters PaLM on 8 out of 8 SuperGLUE tasks and the average SuperGLUE score from 80.7 to 83.1.

- We demonstrate that FCM is scalable – it consistently outperforms PaLM with various model sizes, from 128 million parameters to 1 billion and 8 billion.

## 2 RELATED WORK

**Masking strategies and pre-training objectives.** Many self-supervised pre-training techniques have been proposed to leverage the vast availability of unsupervised data. Different architectures typically leverage different objectives. Decoder-only models are typically trained with causal language model objectives to mimic auto-regressive generation (Brown et al., 2020) and is found to be effective in cross-modality learning (Alayrac et al., 2022; Yu et al., 2022). Related to our masking out tokens, scheduled sampling (Bengio et al., 2015) applied replacing tokens with model predicted tokens and is shown to improve training recurrent neural networks. Autoencoding denoising objectives have been used to learn a bidirectional contextualized encoder for natural language understanding (Devlin et al., 2018; Liu et al., 2019; Yang et al., 2019). For encoder-decoder models, BART (Lewis et al., 2019) conducts NLU tasks by feeding the same input into the encoder and decoder, and taking the final hidden states of the decoder. Raffel et al. (2020) explored many objectives of pre-training and found that span-corruption works best with encoder-decoder model. Other work explores multi-task pre-training using supervised data (Aribandi et al., 2021; Sanh et al., 2022; Wang et al., 2022). To study the impact of different objectives on zero-shot generalization, Wang et al. (2022) conducts a systematic study of different architectures combined with three different pre-training objectives, and found causal language modeling to be effective at zero-shot learning.

**Combining causal and masked language modeling.** There has been work explore training model with multiple objectives to combine causal and masked language modeling under the masked language modeling objective with different attention masks (Dong et al., 2019; Bao et al., 2020). Later work proposes to use blank infilling (Raffel et al., 2020) to randomly blank out continuous spans of tokens from the input text and train the model to sequentially reconstruct the spans (Du et al., 2022). XLNet (Yang et al., 2019) modifies the attention mask in a standard transformer to enable token generation in any permutation of tokens. XLNet uses a two-stream self-attention mechanism, instead of the right-shift, to avoid information leakage in Transformers, but doubles the time cost of pre-training. UL2 (Tay et al., 2022) further proposes to train language model using a mixture of denoisers to combines diverse pre-training paradigms together. Other work explored masking some spans that are predicted at the end of the sequence for bidirectional models (Artetxe et al., 2022) or left-to-right autoregressive models (Aghajanyan et al., 2022; Zhu et al., 2019; Donahue et al., 2020; Fried et al., 2022). Notably, Bavarian et al. (2022) explores moving text in the middle to the

end and predict it autoregressively. Related to our layer-wise attention masking, word masking has been explored in the context of recurrent neural networks (Dai & Le, 2015; Bowman et al., 2015; Xie et al., 2017). Different from prior work, we focus on efficiently improving causal transformer model with masking language modeling, our method does not require complex implementations to change input or output prediction, making it simple to implement and our method does not add extra computation or parameters.

## 3 METHOD

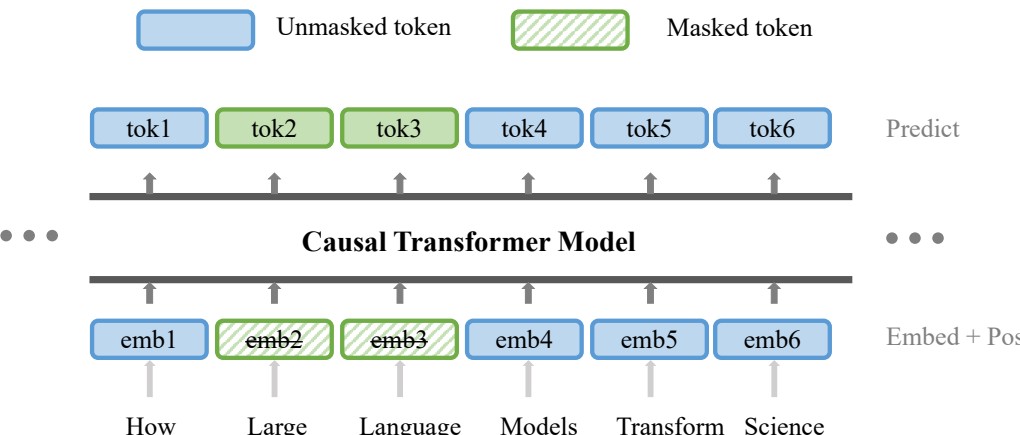

Figure 2: Illustrations of FCM. Given a causal language model, each token's prediction is conditioned on embeddings that are not masked. The loss is applied to each token in the sequence. In this example, when predicting the token of the word *"models"*, the embeddings of the words *"large"* and *"language"* are removed from the input sequence. The model is asked to predict all tokens autoregressively.

### 3.1 PRE-TRAINING OBJECTIVE

FCM uses a standard causal, decoder-only Transformer model architecture (Vaswani et al., 2017), *i.e.*, each timestep can only attend to itself and past timesteps. We illustrate FCM in Figure 2. Given an input text $\mathbf{x} = [x_1, \cdots, x_n]$, the standard causal language modeling objective is defined to maximize the log likelihood of $x$ autoregressively:

$$\log p(\mathbf{x}) = \log \prod_{i=1}^{n} p(x_i | x_1, x_2, \dots, x_{i-1})$$

$$= \log \prod_{i=1}^{n} p(x_i | \mathbf{x}_{<i}) := \log \prod_{i=1}^{n} p(x_i | [x_j]_{j=0}^{i-1}). \quad (1)$$

In FCM, we randomly sample a mask ratio from $m \sim [0, \eta]$ where $\eta \in [0, 1]$ is a fixed maximum mask ratio, we use $\eta = 0.15$ throughout the experiments unless otherwise mentioned. The model is asked to predict each token $x_i \in \mathbf{x}$, and can only attend to tokens in $\mathbf{x}_{<i}$ that are not sampled. Concretely, the FCM objective is given by:

$$\log p(\mathbf{x}) = \log \prod_{i=1}^{n} p(x_i | [I[m_j > \eta] \cdot x_j]_{j=0}^{i-1}), \quad (2)$$

where $m_j \sim \mathcal{U}(0, 1)$. This can be efficiently implemented by combining it with causal attention mask. While applying random masking to the token sequence, we always exclude the special BOS ('beginning of sentence') token at the beginning of each sequence, so that the model is aware of the beginning of a sentence. Moreover, keeping the BOS token unmasked helps with training stability because it ensures that there is at least one token unmasked without changing the semantic

meaning of the sequence. For example, when predicting token $x_t$ for small $t$, it is possible that all tokens $[x_1, ..., x_{t-1}]$ are masked, which can cause instability in the training loss. We found that this technique enables us to train arbitrary high mask ratios without incurring instability.

## 3.2 MODEL ARCHITECTURE

We use the same model and architecture as PaLM (Chowdhery et al., 2022), including the modified activation (Shazeer, 2020), multi-query attention (Shazeer, 2019), parallel layers (Wang & Komatsuzaki, 2021) and RoPE embeddings (Su et al., 2021) described therein, with the exception that we use SentencePiece (Kudo & Richardson, 2018) vocabulary with 32K tokens from C4 (Raffel et al., 2020). To study the dependence of FCM on model size, we train 3 different sizes of the model, ranging over three orders of magnitude from 125 million parameters, to 1 billion parameters, and to 8 billion parameters (see Table 1).

| Model | Layers | # of heads | $d_{model}$ | Batch size | Seq len |
|---|---|---|---|---|---|
| PaLM, FCM 128M | 8 | 4 | 1024 | 1024 | 1024 |
| PaLM, FCM 1B | 16 | 8 | 2048 | 1024 | 1024 |
| PaLM, FCM 8B | 32 | 16 | 4096 | 1024 | 1024 |

Table 1: Architecture details of different sized models. We list the number of layers, $d_{model}$, the number of attention heads and attention head size, training batch size, and sequence length. The feed-forward size $d_{ff}$ is always $4 \times d_{model}$ and attention head size is always 256.

## 3.3 METHOD DISCUSSION

In this section, we discuss the connections and differences between FCM and other pre-training models. We are mainly concerned with how they improve few-shot, zero-shot, and finetuning performance.

**T5** Raffel et al. (2020) and **UL2** (Tay et al., 2022) propose to train encoder-decoder or prefix language model architecture using the span-corruption objective. T5 and UL2 always predict spans in a fixed left-to-right order, and are therefore related to FCM in that our method also predicts masked tokens from left to right. However, FCM is an autoregressive model without an encoder that encodes full context information, so in principle, FCM can be combined together *e.g.* with a prefix language model. Empirically, FCM outperforms T5 and UL2 on NLU tasks with smaller models (1B vs 8B) and fewer number of tokens (180B vs 1000B).

**XLNet** (Yang et al., 2019) is also pre-trained with autoregressive objectives, but there are important distinctions between FCM and XLNet. FCM does not need permutation of the input sequence and designing two-stream self-attention mechanism to avoid the information leak within Transformer, which doubles the time cost of pre-training. Our method is much simpler and more scalable.

**GLM** (Du et al., 2022) proposes to extend autoregressive modeling with bidirectional context. They achieve this by selecting spans and move them to the end of sequence, then unselected tokens and past spans use non-causal attention and tokens within each span use causal attention, similar to a PrefixLM and UL2 (Tay et al., 2022; Raffel et al., 2020).

**UniLM** (Dong et al., 2019; Bao et al., 2020) combines different training objectives together by using different self-attention masks to control the access to context for each token. Similar to BERT and T5, UniLM is trained with an autoencoding objective with masked spans replaced by mask tokens. This introduces a gap between pre-training and downstream tasks, since there are no mask tokens in downstream tasks. Moreover, the model needs to be finetuned for natural language generation tasks (*e.g.*, summarization). In contrast, FCM focuses on improving causal language models and outpeforms strong baselines such UL2 and PaLM on zero- and few-shot SuperGLUE benchmark.

In summary, our method is simpler and focuses on autoregressive causal language models. Our method is easy to implement and does not introduce extra computation or parameters, and as experimental evaluations in Section 4 show, FCM is scalable and achieves superior results than baselines.

## 4 EXPERIMENTS

### 4.1 EXPERIMENTAL SETUP

**Training datasets** We use C4 dataset to pre-train baselines and our model (Raffel et al., 2020). It is a colossal, cleaned version of Common Crawl's web crawl corpus[1] and consists of about 180 billion tokens using sentencepiece tokenizer (Kudo & Richardson, 2018). Note that GPT-3 and PaLM use significantly larger pre-training datasets, *e.g.*, the PaLM pre-training dataset consists of a high-quality corpus of 780 billion tokens that is a mixture of filtered webpages, social media conversations, and more. However, these datasets are not publicly available and training on it requires tremendous compute resources. C4 is significantly smaller, which also reduces the compute cost of training the large models.

**Training and inference.** Our training optimizer follows PaLM, and use the Adafactor optimizer (Shazeer & Stern, 2018) which scales the learning rate by the root-mean-square of the parameter matrix. We use learning rate of $0.01$ for the first 10,000 steps, which is then decayed at a rate of $1/\sqrt{k}$, where $k$ is the step number. We train with momentum of $\beta_1 = 0.9$. The second-order moment interpolation value is computed as $\beta_2 = 1.0 - k^{-0.8}$, where $k$ is the step number. Following typical large Transformer models training as in PaLM and GPT-3, models are trained without dropout, and dropout of 0.1 is used for finetuning. Our training and inference codebase is based on JAX and T5X, and all models are trained on TPU v4 Pods. The few-shot and zero-shot results are averaged over three evaluation random seeds. For results of baselines, we choose and report the best published results to compare against FCM. We use exactly the same batch size, learning rate, and training hyperparameters for PaLM and FCM. More details on hyperparameters, compute infrastructure, and training time are provided in Appendix A.

**Evaluation tasks and metrics.** We consider the following tasks and categorize them according to their focused evaluation properties:

- *Cloze and Completion tasks*: **LAMBADA** (Paperno et al., 2016) consists of word prediction tasks that test the understanding of narrative passages. **StoryCloze** (Mostafazadeh et al., 2016) evaluates story understanding and script understanding, by requiring a system to choose the correct ending to a four-sentence story.

- *Commonsense Reasoning*: **PIQA** (Bisk et al., 2019) is a dataset designed for physical commonsense reasoning to investigate the physical knowledge of language models. **ARC** (Yadav et al., 2019) is a multiple-choice question-answering dataset, containing questions from science exams from grades 3-9. There are two partitioned datasets ARC-e (easy) and ARC-c (challenge), where the latter partition contains the more difficult questions that require reasoning. **OpenBookQA** (Mihaylov et al., 2018) is designed to test understanding of both the topic (*e.g.*, salient facts) and the language it is expressed in. This dataset contains questions that require multi-step reasoning, commonsense knowledge, and rich text comprehension.

- *Winograd-style tasks*: In the Winograd schema challenge, a *schema* is a pair of sentences that differ in only one or two words and that contain an ambiguity that is resolved in opposite ways in the two sentences. **Winograd** tasks (Kocijan et al., 2020) require world knowledge and reasoning to be solved. **WinoGrande** (Sakaguchi et al., 2020) is a large-scale dataset of 44k problems, and requires commonsense reasoning to choose the correct option for a given sentence.

- *Natural Language Understanding (NLU)*: **SuperGLUE** (Sarlin et al., 2020) consists of 8 challenging NLU tasks, including word sense disambiguation, natural language inference, coreference resolution, and question-answering.

- *Natural Language Inference* (NLI): **Adversarial NLI (ANIL)** (Nie et al., 2019) is collected via an adversarial human-and-model-in-the-loop procedure and is selected to be difficult to state-of-the-art models.

**Baselines.** The main baseline we compare with is PaLM (Chowdhery et al., 2022), since it is one of state-of-the-arts on a wide range of NLP benchmarks.

---

[1]https://commoncrawl.org

## 4.2 MAIN RESULTS

### 4.2.1 FEW-SHOT PERFORMANCE

We compare FCM with PaLM on few-shot and zero-shot performance in a wide range of benchmarks. Table 2 includes the results for the FCM and the PaLM 1B and 8B models. The results averaged over task categories are presented in Figure 1. Following prior work, we only consider single checkpoint results from pre-trained language models.

FCM outperforms PaLM on 17 out of 19 tasks in the zero-shot setting, 15 out of 19 tasks in the one-shot setting, and 15 out of 19 tasks in the few-shot setting. On the SuperGLUE (Sarlin et al., 2020) benchmark, our method significantly improves the 1B-model-size PaLM's zero-shot performance from 55.7 to 59.2 and improves the 8B-model-size PaLM's zero-shot performance from 61.6 to 64.0. Consider that PaLM is well-tuned in many aspects, including the pre-training dataset, training strategy, and the number of tokens observed. The significantly better results of FCM shows that the training objective can also play a crucial role in the model performance.

Table 2: Results obtained by the FCM 1B and 8B model across NLP benchmarks. We use the same setup as in Brown et al. (2020); Chowdhery et al. (2022), including the splits for each task.

| Task | Zero-shot | | | | One-shot | | | | Few-shot | | | |
|---|---|---|---|---|---|---|---|---|---|---|---|---|
| | PaLM 1B | FCM 1B | PaLM 8B | FCM 8B | PaLM 1B | FCM 1B | PaLM 8B | FCM 8B | PaLM 1B | FCM 1B | PaLM 8B | FCM 8B |
| Lambada (EM) | 42.4 | **43.5** | 58.0 | **59.1** | 48.9 | **49.5** | 65.8 | **66.5** | 48.2 | **49.7** | 66.1 | **67.5** |
| StoryCloze | **68.8** | 68.2 | 75.0 | **75.6** | **67.3** | 66.9 | 75.0 | **75.7** | 65.9 | **66.7** | 75.8 | **76.2** |
| PIQA | 72.0 | **72.1** | 77.0 | **77.4** | 71.0 | **71.6** | 75.5 | **76.5** | **72.0** | 71.6 | 77.1 | **77.3** |
| ARC-e | **46.2** | 45.6 | 55.3 | **57.1** | **48.0** | 45.9 | 60.1 | **60.2** | **50.2** | 48.2 | 64.0 | **64.4** |
| ARC-c | 25.8 | **27.7** | **33.8** | 33.0 | 26.3 | **27.2** | 34.0 | **35.0** | 26.5 | **28.1** | 35.5 | **36.5** |
| OpenbookQA | 45.8 | **46.4** | 48.2 | **49.2** | **45.0** | 43.2 | 47.0 | **48.4** | 42.6 | **43.6** | 49.0 | **49.5** |
| Winograd | 67.0 | **70.0** | 78.5 | **80.6** | 67.0 | **67.4** | 79.5 | **81.7** | 64.8 | **70.0** | 79.5 | **81.2** |
| Winogrande | 54.0 | **54.5** | 60.0 | **61.9** | 54.0 | **55.8** | 60.5 | **62.1** | 53.6 | **55.0** | 61.0 | **62.3** |
| BoolQ | 45.9 | **56.0** | 52.0 | **62.1** | 48.3 | **52.6** | 53.7 | **59.6** | **48.1** | 46.8 | 49.0 | **57.7** |
| Copa | 72.0 | **74.0** | 82.0 | **84.0** | 72.0 | **73.0** | 80.0 | **83.0** | 70.0 | **72.0** | 82.0 | **85.0** |
| RTE | 50.9 | **53.8** | **53.4** | 48.9 | 53.1 | **54.5** | **55.2** | 47.3 | **53.1** | 45.1 | **53.1** | 48.4 |
| WiC | 51.4 | **52.6** | 78.3 | **79.1** | **47.8** | 46.9 | 79.0 | **86.8** | 48.9 | **50.1** | 77.9 | **87.9** |
| Multirc (F1a) | 35.2 | **40.6** | 40.4 | **54.1** | 57.1 | **57.2** | 49.8 | **56.5** | **57.2** | 48.2 | 42.5 | **46.5** |
| WSC | 65.3 | **70.2** | 78.3 | **79.1** | 66.7 | **71.2** | 79.0 | **86.8** | 66.7 | **70.2** | 77.9 | **87.9** |
| ReCoRD | 75.8 | **76.3** | **85.5** | 85.0 | 75.8 | **76.4** | **85.5** | 84.9 | 74.9 | **75.0** | **84.6** | 83.9 |
| CB | 48.2 | **50.0** | 82.0 | **84.0** | 44.6 | **44.8** | 42.9 | **51.5** | 42.3 | **48.2** | 46.4 | **50.0** |
| ANLI R1 | 33.3 | **33.5** | 32.9 | **34.3** | 31.3 | **33.0** | 32.7 | **33.5** | 30.5 | **32.5** | 31.1 | **32.9** |
| ANLI R2 | 32.8 | **34.2** | 33.3 | **34.1** | 30.5 | **30.6** | 30.6 | **33.7** | 32.5 | **33.4** | 31.7 | **33.8** |
| ANLI R3 | 33.3 | **33.6** | 33.0 | **33.9** | 30.0 | **31.2** | 31.7 | **33.8** | 32.8 | **34.2** | 32.9 | **35.1** |

### 4.2.2 FINETUNING PERFORMANCE

We conduct finetuning experiments on the SuperGLUE benchmark to compare PaLM and FCM. Following PaLM experimental settings, models are finetuned with $5 \times 10^{-5}$ learning rate using the Adafactor optimizer. To reduce computation time, we use batch size 512 instead of the original batch size 32 in PaLM. The models are finetuned for 20K steps.

Table 3 reports the *validation* results on finetuning on task-proportionate mixture of SuperGLUE tasks. On SuperGLUE, we compare with state-of-the-art models such as T5 11B (Raffel et al., 2020) and UL2 (Tay et al., 2022), as well as PaLM (Chowdhery et al., 2022) and show that FCM obtains significantly better performance than PaLM. All models are trained on C4 dataset, T5 11B and UL2 are trained on 1000B tokens, the rest of models are trained on 180B tokens. It is worth noting that both top performing models on SuperGLUE are encoder-decoder models that are trained using the span-corruption objective. It has been shown that such an architecture generally outperforms autoregressive decoder-only models on classification task finetuning, when training cost is equalized (Raffel et al., 2020). These results demonstrate that FCM can help bridge the gap. FCM

1B outperforms PaLM 1B significantly on 4 out of 8 SuperGLUE tasks, and FCM 8B significantly outperforms PaLM 8B on all 8 SuperGLUE tasks, improving the score from 80.7 to 83.1.

Table 3: Finetuning results on SuperGLUE dev set. We compare with T5-11B (Raffel et al., 2020), UL2 (Tay et al., 2022) and PaLM (Chowdhery et al., 2022). Scores reported are the peak validation scores per task following the setup of T5. All models are trained on the same 180B tokens except that UL2 20B and T5 11B are trained on 1000B tokens.

| Model | BoolQ | CB | CoPA | MultiRC | Record | RTE | WiC | WSC | Avg |
|---|---|---|---|---|---|---|---|---|---|
| Masked language model | | | | | | | | | |
| T5 11B | 90.8 | 94.9 | 98.0 | 87.4 | 93.8 | 93.9 | 77.3 | 96.2 | 89.9 |
| UL2 20B | 90.8 | 98.7 | 99.0 | 88.4 | 93.7 | 92.1 | 77.3 | 98.1 | **90.7** |
| T5 1.4B | 83.7 | 92.9 | 85.9 | 82.7 | 69.6 | 78 | 80.8 | 80 | 81.7 |
| Causal language model | | | | | | | | | |
| PaLM 1B | 75.0 | 78.6 | 58.2 | 62.9 | 63.3 | 60.0 | 66.4 | 71.2 | 67.0 |
| FCM 1B | 73.5 | 82.0 | 58.0 | 63.0 | 65.1 | 61.0 | 66.5 | 80.8 | **68.7** |
| PaLM 8B | 83.7 | 94.6 | 80 | 81 | 71.2 | 80 | 75.2 | 80.1 | 80.7 |
| FCM 8B | 84.8 | 96.4 | 81 | 82.1 | 73.7 | 86 | 76.2 | 85 | **83.1** |

### 4.2.3 MODEL SCALABILITY

To demonstrate the scalability of FCM, we further evaluate FCM with different model sizes in Table 4. We consider both a smaller model with 128 millions parameters, and a scaled-up model with 8 billion parameters. All models are trained for 180 billion tokens, which is equivalent to about one epoch on the C4 dataset. Although 8 billion model size is relatively small compared to typical large language models (Chowdhery et al., 2022; Brown et al., 2020), these results serve as a proof-of-concept of FCM's effectiveness and scalability for larger model sizes. We leave further scaled-up experiments as promising future work. We compare with methods that use more high quality datasets including the official PaLM* and GPT-3. PaLM is trained on 780 billion tokens and GPT-3 is trained on 300 billion tokens. Other baselines include T5 and UL2 which are trained for 1000 billion tokens and ST-MoE which uses 500 billion tokens. The results show that FCM works with both smaller and larger models, ranging from 128 million parameters to 8 billion parameters. Surprisingly, the largest FCM model matches the score of PaLM* with an average score of 64, despite the fact that PaLM* is trained on a much larger dataset (780B tokens) of high-quality conversation and webpage data, while FCM is trained on the smaller C4 dataset (180B tokens). We further compare FCM with official PaLM 8B model on one-shot and few-shot experiments. Table 5 shows the results, FCM matches PaLM in most tasks, showing the promising capabilities of FCM.

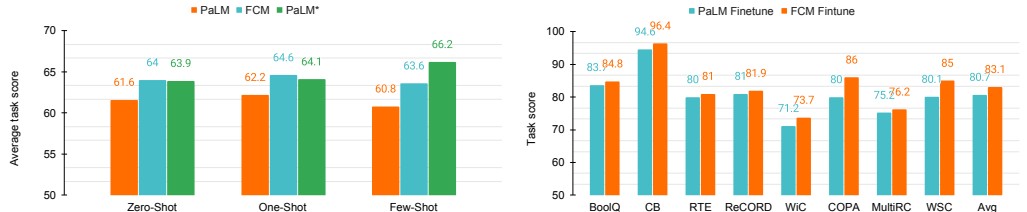

Figure 3: Fewshot and finetuning results on SuperGLUE. We compare PaLM* which is trained on 780B high quality data, PaLM trained on C4, and FCM. On zero-shot and one-shot learning, FCM matches or outperforms PaLM* and outperforms PaLM in few-shot and finetuning significantly.

### 4.3 ABLATION STUDY

**FCM works best with random ratio.** We evaluate the impact of mask ratio on FCM using SuperGLUE zero-shot benchmark. Table 6 presents the results of FCM with different mask ratios. Among them, sampling random ratio between $[0.0, 0.15]$ performs significantly better than other choices. Sampling mask ratios from 0.0 to 0.1 or 0.15 perform generally better than using fixed

Table 4: Comparisons on SuperGLUE zero-shot benchmark with different model sizes. PaLM$^\star$ denotes the published results of PaLM.

| Model | BoolQ | CB | COPA | MultiRC | ReCORD | RTE | WiC | WSC | Avg |
|---|---|---|---|---|---|---|---|---|---|
| Methods that use more high quality data | | | | | | | | | |
| GPT-3 175B | 60.5 | 46.4 | 91 | 72.9 | 90.2 | 63.5 | 0 | 65.4 | 61.2 |
| PaLM$^\star$ 540B | 88 | 51.8 | 93 | 83.5 | 92.9 | 72.9 | 59.1 | 89.1 | 78.8 |
| GPT-3 13B | 66.2 | 19.6 | 84 | 71.4 | 89 | 62.8 | 0 | 64.4 | 57.2 |
| PaLM$^\star$ 8B | 68.3 | 41.1 | 86 | 47.5 | 87.8 | 54.2 | 47 | 78.9 | **63.9** |
| Methods that use 1000B tokens from C4 | | | | | | | | | |
| ST-MoE 269B | 40.8 | 41.1 | 56 | 30.3 | 50 | 52.7 | 50 | 57.5 | 47.3 |
| T5-XXL 11B | 44.3 | 37.5 | 70 | 23 | 85.8 | 48.8 | 50.9 | 59.3 | 52.5 |
| UL2 20B | 63.1 | 41.1 | 85 | 36.2 | 88.1 | 60.7 | 49.8 | 79.9 | **63** |
| Methods that use 180B tokens from C4 | | | | | | | | | |
| PaLM 128M | 58.8 | 8.8 | 63 | 54.3 | 62.4 | 53.1 | 49 | 56.5 | 50.7 |
| FCM 128M | 55.7 | 8.9 | 69 | 55.1 | 62.1 | 56.3 | 52.1 | 56.1 | 51.9 |
| PaLM 1B | 45.9 | 48.2 | 72 | 35.2 | 75.8 | 50.9 | 51.6 | 65.3 | 55.6 |
| FCM 1B | 56 | 50 | 74 | 40.6 | 76.3 | 53.8 | 52.4 | 70.2 | 59.2 |
| PaLM 8B | 52 | 50 | 82 | 40.4 | 85.5 | 53.4 | 51.3 | 78.3 | 61.6 |
| FCM 8B | 62.1 | 48.2 | 84 | 54.1 | 85 | 48 | 51.1 | 79.1 | **64.0** |

Table 5: Comparison between FCM and PaLM on SuperGLUE zero-shot and few-shot benchmark tasks. PaLM$^\star$ denotes published results obtained by training on more high quality data. The model size is 8B.

| # of shots | Model | BoolQ | CB | COPA | MultiRC | ReCORD | RTE | WiC | WSC | Avg |
|---|---|---|---|---|---|---|---|---|---|---|
| Zero-shot | PaLM$^\star$ 8B | 68.3 | 41.1 | 86 | 47.5 | 87.8 | 54.2 | 47 | 78.9 | 63.9 |
| | PaLM 8B | 52 | **50** | 82 | 40.4 | **85.5** | **53.4** | **51.3** | 78.3 | 61.6 |
| | FCM 8B | **62.1** | 48.2 | **84** | **54.1** | 85 | 48 | 51.1 | **79.1** | **64** |
| One-shot | PaLM$^\star$ 8B | 64.7 | 41.1 | 82 | 50.6 | 87.8 | 57.8 | 47.3 | 81.4 | 64.1 |
| | PaLM 8B | 53.7 | 42.9 | 80 | 49.8 | **85.5** | **55.2** | **51.5** | 79 | 62.2 |
| | FCM 8B | **59.6** | **51.5** | **83** | **56.5** | 84.9 | 47.3 | 46.9 | **86.8** | **64.6** |
| Few-shot | PaLM$^\star$ 8B | 68.9 | 57.1 | 82 | 41.1 | 88 | 56.7 | 52.4 | 83.2 | 66.2 |
| | PaLM 8B | 49 | 46.4 | 82 | 42.5 | **84.6** | **53.1** | **50.5** | 77.9 | 60.8 |
| | FCM 8B | **57.7** | **50** | **85** | **46.5** | 83.9 | 48.4 | 49.5 | **87.9** | 63.6 |

mask ratio 0.1 or 0.15, indicating that fixed mask ratios could potentially introduce pre-training and inference gap, and sampling random mask ratio is a simple way to alleviate it.

**Using mask tokens instead of attention mask.** Alternative to FCM, a natural way of preventing future tokens from attending to past tokens is replacing tokens with a special `[mask]` token. Using mask tokens is widely adapted in masked language modeling (Devlin et al., 2018; Liu et al., 2019), and combining mask token with causal language modeling can be considered as a special case of UniLM Dong et al. (2019). We perform an ablation study comparing FCM with mask token, and present the results in Table 7. Using mask tokens lead to performance degradation in zero- and few-shot experiments, and about the same results on finetuning experiments. We hypothesis that the performance drop is due to the train and inference gap caused by introducing the `[mask]` token, which can negatively impact zero- and few-shot performance because the model is not finetuned to remove such gap.

**Comparison with dropout.** FCM random masking can be seen as a special type of dropout (Srivastava et al., 2014) applied only on the input sequence layer wisely by using attention masking. We note that general dropout and FCM are complementary in that they can be combined together. To compare random masking vs. dropout, we compare three models in Table 8: (1) PaLM, (2) PaLM

Table 6: Ablation of mask ratio on SuperGLUE. Comparisons on SuperGLUE zero-shot and one-shot benchmark between fixed mask ratio and random mask ratios using FCM. The model size is 1B. FCM $[x, y]$ denotes mask ratio is randomly sampled between $x$ and $y$.

| Model | BoolQ | CB | COPA | MultiRC | ReCORD | RTE | WiC | WSC | Avg |
|---|---|---|---|---|---|---|---|---|---|
| | | | | zero-shot | | | | | |
| PaLM | 45.9 | 48.2 | 72.4 | 35.2 | 75.8 | 50.9 | 51.6 | 65.3 | 55.7 |
| FCM [0.1, 0.1] | 56.5 | 51.6 | 73.5 | 32.9 | 76.3 | **55.6** | 52 | 67.1 | 58.2 |
| FCM [0.15, 0.15] | 54 | 48.2 | 75.5 | 22.6 | 75.9 | 52.7 | 49.8 | 66.1 | 55.6 |
| FCM [0, 0.1] | **57.9** | 51.8 | 69.6 | 33.3 | **76.8** | 48.4 | 51.6 | 67.7 | 57.1 |
| FCM [0.0, 0.15] | 56 | 50 | **74.1** | 40.6 | 76.3 | 53.8 | **52.4** | **70.2** | **59.2** |
| FCM [0.0, 0.3] | 52.5 | **53.6** | 69.4 | **42.9** | 75.4 | 49.8 | 48.4 | 66.1 | 57.3 |
| | | | | one-shot | | | | | |
| PaLM | 48.3 | 44.6 | 50.9 | 75.8 | 47.8 | 72 | 35.9 | 66.7 | 55.3 |
| FCM [0.1, 0.1] | 56.1 | 32.1 | 53.8 | 76.3 | 47.3 | 72 | 33.3 | 67.4 | 54.8 |
| FCM [0.15, 0.15] | 48.3 | 37.5 | 52.4 | 75.9 | **49.1** | 69 | 20.9 | 66.3 | 52.4 |
| FCM [0, 0.1] | **56.5** | 42.9 | 53.1 | **76.8** | 47.8 | 72 | 29.1 | 66.3 | 55.6 |
| FCM [0.0, 0.15] | 52.6 | **44.6** | **54.5** | 76.4 | 46.9 | 73 | 43.2 | **71.6** | **57.9** |
| FCM [0.0, 0.3] | 50.8 | 32.1 | 52 | 75.4 | 47.2 | **74** | **46.5** | 66.3 | 55.5 |

Table 7: Comparisons on SuperGLUE zero-shot, few-shot and finetuning benchmarks between using attention mask and using mask token. The model size is 1B.

| Masking strategy | 0-shot avg | 1-shot avg | 5-shot avg | finetune avg |
|---|---|---|---|---|
| `Mask` token | 57.4 | 57.0 | 55.4 | 68.5 |
| Attention | 59.2 | 57.9 | 57.2 | 68.7 |

with dropout rate 0.1, (3) FCM with fixed random ratio 0.1, and (4) FCM with fixed random ratio 0.1 and dropout rate 0.1. We see that using dropout during large language models pre-training is harmful, decreasing the score from 55.7 to 55.4, which aligns with findings from prior work (Raffel et al., 2020; Chowdhery et al., 2022). In contrast, combining dropout with FCM together can improve PaLM, improving the score 55.7 to 57.7, indicating that FCM and dropout are complementary techniques and we leave further studies of this as an interesting future work. We can see that using only FCM performs slightly better than combining dropout and FCM together, showing the effectiveness of FCM on performing the next token prediction task with randomly selected past tokens masked out.

Table 8: Comparisons on SuperGLUE zero-shot benchmark between between Random Masking vs. Dropout. The model size is 1B.

| Model | BoolQ | CB | COPA | MultiRC | ReCORD | RTE | WiC | WSC | Avg |
|---|---|---|---|---|---|---|---|---|---|
| PaLM | 45.9 | 48.2 | 72.4 | 35.2 | 75.8 | 50.9 | 51.6 | 65.3 | 55.7 |
| PaLM + Dropout | 53.5 | 48.2 | 64.4 | 37.2 | 75.7 | 50.2 | 50.2 | 63.5 | 55.4 |
| FCM [0.1, 0.1] + Dropout | 44 | **53.6** | 71 | **43.1** | 75.3 | **59.2** | 49.8 | 65.4 | 57.7 |
| FCM [0.1, 0.1] | **56.5** | 51.6 | **73.5** | 32.9 | **76.3** | 55.6 | **52** | **67.1** | **58.2** |

## 5 CONCLUSION

In this paper, we propose FCM, a novel pre-training paradigm using a causal transformer decoder. FCM is a combination of causal next-token-prediction and random masking to input sequence. Experimental results show that FCM significantly outperforms the state-of-the-art causal transformer model on a wide range of zero- and few-shot as well as finetuning benchmarks, and our model is readily extendable to various tasks.

As FCM improves performance of causal language models on few-shot and finetuning benchmarks, applying our approach to other language understanding tasks and language-image tasks (*e.g.*, Flamingo (Alayrac et al., 2022)) is a promising direction for future work. Since our method does not introduce extra computation, another direction would be investigating what is the impact of FCM on compute scaling law (Hoffmann et al., 2022).

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

## A APPENDIX

### A.1 IMPLEMENTATION AND TRAINING DETAILS

Our implementation uses Flax (Heek et al., 2020), JAX (Bradbury et al., 2018) and T5X (Roberts et al., 2022) Our architecture is based on PaLM (Chowdhery et al., 2022) which introduces some modifications to GPT-3 (Brown et al., 2020) architecture to reduce compute cost. The dataset C4 is provided by Tensorflow datasets. We use SentencePiece (Kudo & Richardson, 2018) as tokenizer. For PaLM and FCM trained on C4 datasets, the sequence length is 1024 to reduce compute cost, although the official PaLM is trained with longer context length 2048. Following the settings of PaLM, input examples are concatenated together and then split into sequences of exactly 1024 tokens, so that there are no padding tokens, but examples may be split in the middle. Input examples are differentiated from one another with a special [eod] token For downstream tasks evaluation, including both fewshot and finetune benchmarks, we follow the dataset format and splits used in Brown et al. (2020); Chowdhery et al. (2022). Our experiments are conducted using cloud TPU v4, it has a unified 32 GiB HBM memory space across the entire chip. For 128M models, training 180B tokens takes 18 hours on TPU v4-64. For 1B models, training on 180B tokens takes 25 hours on TPU v4-256. The training of 8B models on 180B tokens takes 100 hours on TPU v4-512.

### A.2 HYPERPARAMETERS

In this section we provide the training and evaluation hyperparameters of FCM. These configurations follow the training hyperparameters of PaLM Chowdhery et al. (2022).

| Hyperparameter | Value |
|---|---|
| Dropout | 0.0 |
| Optimizer | Adafactor |
| Initial learning rate $lr$ | 0.01 |
| Learning rate decay | 0.01 for the first 10,000 steps, which is then decayed at a rate of $1/\sqrt{k}$, where $k$ is the step number |
| Weight decay | $lr^2$ |
| Optimizer momentum | $\beta_1 = 0.9, \beta_2 = 1.0 - k^{-0.8}$ |
| Global norm gradient clipping | 1.0 |
| Batch size | 1024 |
| Sequence length | 1024 |

Table 9: Hyperparameters for training PaLM and FCM

| Hyperparameter | Value |
|---|---|
| Dropout | 0.1 |
| Optimizer | SGD momentum |
| Momentum | 0.9 |
| Batch size | 512 |
| Sequence length | 1024 |

Table 10: Hyperparameters for finetuning PaLM and FCM

### A.3 FULL RESULTS

Table 12 includes the results for the FCM and the PaLM 1B and 8B models across three random evaluation seeds. Following prior work, we only consider single checkpoint results from pre-trained language models. The variance across different evaluation seeds is small on most tasks.

### A.4 ADDITIONAL EXPERIMENTS ON COMPARING WITH FILL-IN-THE-MIDDLE

Related to FCM is fill-in-the-middle training, which splits documents into three pieces at random and moves the middle piece to the end. This is similar to the procedure used in Aghajanyan et al. (2022); Donahue et al. (2020); Bavarian et al. (2022); Artetxe et al. (2022). FCM is orthogonal to fill-in-the-middle training, and in fact, our method can be easily integrated into such methods:

CM3 (Aghajanyan et al., 2022) and HYBUNI (Artetxe et al., 2022) focus on introducing bidirectionality into causal masking objectives, where the masked tokens are moved to the end of the sequence in order to make it possible to attend to future positions from these masked tokens. The introduction of bidirectionality gives the model additional capabilities to attend to the future when predicting masked tokens.

On the contrary, our paper focuses exclusively on the setting of unidirectional language models. Unlike CM3 and HYBUNI, our approach focuses on improving autoregressive language models without altering the input sequence or changing the training objective. Instead, we only change the attention masks for unidirectional language models. In fact, we can incorporate our attention masking method into bidirectional models such as CM3 and HYBUNI, but it is beyond the scope of the paper and we leave it for future work.

In Table 11, we compare with a simpler version of CM3 and HYBUNI, where we randomly split the sentence into [prefix, middle, suffix], move [middle] to the end of the sequence, concatenate the three pieces using sentinel tokens, and train the causal language model to predict the sequence. We report zero- and few-shot results on SuperGLUE. The fill-in-the-middle is denoted as FIM. We did not see an improvement in zero- and few-shot learning. Bavarian et al. (2022) also reported a similar finding that few-shot performance does not improve by moving the infill regions to the end of the context.

CM3 demonstrates impressive results on cross modal generation and representation learning tasks, we believe combining CM3 and FCM could further improve the performance which we leave as an interesting future work.

Table 11: Comparison between FCM, PaLM, and fill-in-the-middle (FIM) on SuperGLUE zero-shot and few-shot benchmark tasks. All models are trained for 180B tokens on C4. The model size is 1B.

| # of shots | Model | BoolQ | CB | COPA | MultiRC | ReCORD | RTE | WiC | WSC | Avg |
|---|---|---|---|---|---|---|---|---|---|---|
| Zero-shot | PaLM 1B | 52 | **50** | 82 | 40.4 | **85.5** | **53.4** | **51.3** | 78.3 | 61.6 |
| | FIM 1B | 50.5 | 47.7 | 78 | 39.8 | 83.7 | 47.9 | 50.1 | 77.9 | 59.8 |
| | FCM 1B | **62.1** | 48.2 | **84** | **54.1** | 85 | 48 | 51.1 | **79.1** | **64** |
| | | | | | | | | | | |
| One-shot | PaLM 1B | 53.7 | 42.9 | 80 | 49.8 | **85.5** | **55.2** | **51.5** | 79 | 62.2 |
| | FIM 1B | 52 | 43.1 | 78 | 49 | 83.1 | 48.2 | 45.9 | 80 | 59.9 |
| | FCM 1B | **59.6** | **51.5** | **83** | **56.5** | 84.9 | 47.3 | 46.9 | **86.8** | **64.6** |
| | | | | | | | | | | |
| Few-shot | PaLM 1B | 49 | 46.4 | 82 | 42.5 | **84.6** | **53.1** | **50.5** | 77.9 | 60.8 |
| | FIM 1B | 50.5 | 45.6 | 79 | 39.8 | 80.1 | 50.9 | 49.4 | 75.8 | 58.8 |
| | FCM 1B | **57.7** | **50** | **85** | **46.5** | 83.9 | 48.4 | 49.5 | **87.9** | **63.6** |

Table 12: Results across three random realizations. We use the same setup as in Brown et al. (2020); Chowdhery et al. (2022), including the splits for each task.

| Task | One-shot PaLM 1B | FCM 1B | PaLM 8B | FCM 8B | Few-shot PaLM 1B | FCM 1B | PaLM 8B | FCM 8B |
|---|---|---|---|---|---|---|---|---|
| Lambada (EM) | 48.9 | **49.5** | 65.8 | **66.5** | 48.2 | **49.7** | 66.1 | **67.5** |
|  | 48.6 / 48.9 / 49.3 | 49.4 / 49.4 / 49.6 | 65.8 / 65.8 / 65.9 | 66.6 / 66.5 / 66.5 | 48.2 / 48.2 / 48.2 | 49.6 / 49.7 / 49.7 | 66.1 / 66.0 / 66.1 | 67.5 / 67.5 / 67.4 |
| StoryCloze | **67.3** | 66.9 | 75.0 | **75.7** | 65.9 | **66.7** | 75.8 | **76.2** |
|  | 67.0 / 67.6 / 67.4 | 66.7 / 66.9 / 67.1 | 75.1 / 75.0 / 74.8 | 75.5 / 75.9 / 75.8 | 65.8 / 65.9 / 65.9 | 66.7 / 66.5 / 66.6 | 75.6 / 75.9 / 75.8 | 76.0 / 76.5 / 76.3 |
| PIQA | 71.0 | **71.6** | 75.5 | **76.5** | **72.0** | 71.6 | 77.1 | **77.3** |
|  | 71.0 / 71.0 / 71.0 | 71.7 / 71.6 / 71.6 | 75.5 / 75.4 / 75.6 | 76.5 / 76.5 / 76.5 | 72.0 / 72.1 / 72.1 | 71.6 / 71.6 / 71.7 | 77.1 / 77.1 / 77.1 | 77.3 / 77.3 / 77.3 |
| ARC-e | **48.0** | 45.9 | 60.1 | **60.2** | **50.2** | 48.2 | 64.0 | **64.4** |
|  | 48.0 / 48.0 / 48.1 | 45.9 / 45.9 / 45.9 | 60.2 / 60.1 / 60.1 | 60.2 / 60.3 / 60.3 | 50.2 / 50.2 / 50.2 | 48.2 / 48.1 / 48.2 | 64.0 / 64.0 / 64.0 | 64.0 / 64.8 / 64.5 |
| ARC-c | 26.3 | **27.2** | 34.0 | **35.0** | 26.5 | **28.1** | 35.5 | **36.5** |
|  | 26.3 / 26.6 / 26.0 | 27.1 / 27.4 / 27.3 | 34.0 / 34.0 / 34.1 | 35.1 / 35.0 / 35.0 | 26.1 / 26.9 / 26.6 | 28.1 / 28.1 / 28.1 | 35.5 / 35.5 / 35.5 | 36.5 / 37.1 / 35.9 |
| Openbook-QA | **45.0** | 43.2 | 47.0 | **48.4** | 42.6 | **43.6** | 49.0 | **49.5** |
|  | 45.0 / 45.0 / 45.0 | 43.2 / 43.2 / 43.2 | 47.0 / 47.0 / 47.1 | 48.3 / 48.5 / 45.5 | 42.6 / 42.6 / 42.7 | 43.5 / 43.6 / 43.6 | 49.0 / 49.0 / 49.0 | 49.0 / 49.1 / 50.5 |
| Winograd | 67.0 | **67.4** | 79.5 | **81.7** | 64.8 | **70.0** | 79.5 | **81.2** |
|  | 67.1 / 67.0 / 67.0 | 67.5 / 67.4 / 67.4 | 79.5 / 79.5 / 79.5 | 81.7 / 81.8 / 81.7 | 65.0 / 64.6 / 64.8 | 70.0 / 70.0 / 70.0 | 79.5 / 79.5 / 79.5 | 81.0 / 81.4 / 82.2 |
| Winogrande | 54.0 | **55.8** | 60.5 | **62.1** | 53.6 | **55.0** | 61.0 | **62.3** |
|  | 54.0 / 54.0 / 54.0 | 55.8 / 55.8 / 55.9 | 60.1 / 60.5 / 60.9 | 62.1 / 62.2 / 62.1 | 53.6 / 53.6 / 53.7 | 55.9 / 55.8 / 53.3 | 59.1 / 58.9 / 65.0 | 62.3 / 62.4 / 62.3 |
| BoolQ | 48.3 | **52.6** | 53.7 | **59.6** | **48.1** | 46.8 | 49.0 | **57.7** |
|  | 48.0 / 48.3 / 48.7 | 52.8 / 52.6 / 52.4 | 53.5 / 53.7 / 53.9 | 59.0 / 60.0 / 59.8 | 48.1 / 48.2 / 48.1 | 46.8 / 46.8 / 46.9 | 49.0 / 49.0 / 49.1 | 56.7 / 57.9 / 58.5 |
| Copa | 72.0 | **73.0** | 80.0 | **83.0** | 70.0 | **72.0** | 82.0 | **85.0** |
|  | 71.0 / 73.0 / 74.0 | 73.0 / 73.0 / 73.0 | 79.0 / 80.0 / 81.0 | 82.0 / 84.0 / 83.0 | 70.1 / 71.0 / 70.0 | 72.0 / 71.0 / 73.0 | 83.0 / 81.0 / 82.0 | 85.0 / 85.0 / 85.0 |
| RTE | 53.1 | **54.5** | **55.2** | 47.3 | **53.1** | 45.1 | **53.1** | 48.4 |
|  | 52.0 / 53.1 / 54.2 | 54.5 / 54.5 / 54.5 | 55.0 / 55.2 / 55.4 | 47.3 / 47.3 / 47.3 | 53.0 / 53.1 / 53.1 | 45.1 / 45.1 / 45.1 | 53.1 / 53.1 / 53.1 | 48.4 / 48.4 / 48.3 |
| WiC | **47.8** | 46.9 | 79.0 | **86.8** | 48.9 | **50.1** | 77.9 | **87.9** |
|  | 47.8 / 47.8 / 47.8 | 46.9 / 46.9 / 46.9 | 79.0 / 79.0 / 79.0 | 86.0 / 87.0 / 87.4 | 48.9 / 48.9 / 48.9 | 50.0 / 50.0 / 50.1 | 77.9 / 77.9 / 77.8 | 88.4 / 87.4 / 87.9 |
| Multirc (F1a) | 57.1 | **57.2** | 49.8 | **56.5** | **57.2** | 48.2 | 42.5 | **46.5** |
|  | 57.1 / 57.1 / 57.0 | 57.0 / 57.4 / 57.2 | 49.8 / 49.8 / 49.9 | 56.9 / 56.0 / 56.6 | 57.2 / 57.3 / 57.1 | 48.2 / 47.9 / 48.5 | 42.3 / 42.3 / 42.9 | 46.5 / 46.2 / 46.8 |
| WSC | 66.7 | **71.2** | 79.0 | **86.8** | 66.7 | **70.2** | 77.9 | **87.9** |
|  | 66.5 / 66.5 / 67.1 | 71.5 / 71.2 / 69.9 | 79.0 / 79.0 / 79.0 | 86.8 / 86.9 / 86.7 | 66.7 / 66.7 / 66.7 | 70.0 / 70.5 / 70.1 | 77.9 / 77.9 / 77.9 | 87.8 / 87.8 / 87.8 |
| ReCoRD | 75.8 | **76.4** | **85.5** | 84.9 | 74.9 | **75.0** | **84.6** | 83.9 |
|  | 75.8 / 75.7 / 75.9 | 76.4 / 76.4 / 76.4 | 85.8 / 85.2 / 85.5 | 84.9 / 83.9 / 85.9 | 74.9 / 74.9 / 74.9 | 75.0 / 75.0 / 75.1 | 84.8 / 84.8 / 84.2 | 84.0 / 84.1 / 83.6 |
| CB | 44.6 | **44.8** | 42.9 | **51.5** | 42.3 | **48.2** | 46.4 | **50.0** |
|  | 44.9 / 44.2 / 44.7 |  | 42.9 / 42.9 / 42.9 | 50.5 / 52.0 / 52.0 | 42.3 / 42.3 / 42.3 | 48.1 / 48.1 / 48.4 | 46.6 / 46.6 / 46.0 | 51.0 / 51.0 / 48.0 |
| ANLI R1 | 31.3 | **33.0** | 32.7 | **33.5** | 30.5 | **32.5** | 31.1 | **32.9** |
|  | 31.3 / 31.3 / 31.3 | 33.9 / 33.1 / 32.0 | 32.7 / 32.9 / 32.5 | 33.5 / 33.5 / 33.6 | 30.8 / 30.8 / 29.9 | 32.5 / 32.5 / 32.5 | 31.1 / 31.2 / 31.0 | 32.1 / 32.1 / 32.5 |
| ANLI R2 | 30.5 | **30.6** | 30.6 | **33.7** | 32.5 | **33.4** | 31.7 | **33.8** |
|  | 30.6 / 30.6 / 30.3 | 30.4 / 30.4 / 31.0 | 30.6 / 30.6 / 30.7 | 34.0 / 34.0 / 33.1 | 32.5 / 32.5 / 32.5 | 33.0 / 33.2 / 34.0 | 31.7 / 31.7 / 31.7 | 33.9 / 33.9 / 33.6 |
| ANLI R3 | 30.0 | **31.2** | 31.7 | **33.8** | 32.8 | **34.2** | 32.9 | **35.1** |
|  | 30.2 / 30.2 / 29.6 | 31.0 / 31.1 / 32.5 | 31.5 / 31.5 / 32.1 | 33.5 / 33.5 / 34.4 | 32.6 / 32.6 / 33.2 | 34.0 / 34.0 / 34.6 | 32.8 / 32.7 / 33.2 | 34.1 / 35.9 / 35.3 |

