# OpenReview forum: "Forgetful causal masking makes causal language models better zero-shot learners"
_ICLR.cc/2023/Conference — Submitted to ICLR 2023_

### Official Review · Reviewer_wa4A · 2022-10-24

**Confidence:** 5
**Correctness:** 3
**Technical Novelty And Significance:** 2
**Empirical Novelty And Significance:** 1
**Recommendation:** 3

**Clarity, Quality, Novelty And Reproducibility:**

*Quality*:

Due to lack of comparisons with relevant methods, I don’t think this paper has a high quality.

*Novelty*:

The casual masking method proposed by this paper is quite similar to [2] and HYBUNI in [1]. [2] also adds some masks on the basis of casual language modeling. The difference is that [2] puts the masked tokens to the end of sentence, so the masked places can not only attend to previous tokens but also gather information from future tokens. Compared to the FCM in this paper, I think the performance of causal masking methods from [2] might be better. But due to lack of comparison, I cannot see the performance differences.

*Reproducibility*:
easy to reproduce.

*References*

[1]. Artetxe, Mikel, et al. "On the Role of Bidirectionality in Language Model Pre-Training." arXiv preprint arXiv:2205.11726 (2022).

[2]. Aghajanyan, Armen, et al. "Cm3: A causal masked multimodal model of the internet." arXiv preprint arXiv:2201.07520 (2022).

**Strength And Weaknesses:**

*Strength*:
This paper is well written and easy to understand.

*Weaknesses*:

The biggest weakness of this paper is ignoring related research papers and lack of comparison.
[1] presents comprehensive study about different training objectives in zero-shot priming and finetuning. They compared 6 different training objectives in table 1 (https://arxiv.org/pdf/2205.11726.pdf): NXTUNI, NXTPRE, MSKUNI, MSKBI, HYBUNI, HYBPRE. Casual language modeling such as GPT models belong to NXTUNI. They controlled hyper-parameters and training data to be same and studied these six different objectives. The most relevant method to this paper is HYBUNI (casual masking objective). [2] is the representative work of causal masking objective. This paper presents this new causal masking objective without comparing it with the most relevant casual masking method (CM3).

In addition, this paper should carry out apple-to-apple comparisons with other models. For example, it needs to control model sizes (model parameters), batch size, hyper-parameters, training tokens, training devices, to be exactly the same. And control Total train compute (flops) to be similar. Therefore, the qualified apple-to-apple comparison is only PaLM 1B v.s. FCM 1B (in table 3), PaLM 128M v.s. FCM 128M (in table 4). For others, PaLM 8B used different training data with FCM 8B, and T5 used different model parameters.

*References*

[1]. Artetxe, Mikel, et al. "On the Role of Bidirectionality in Language Model Pre-Training." arXiv preprint arXiv:2205.11726 (2022).

[2]. Aghajanyan, Armen, et al. "Cm3: A causal masked multimodal model of the internet." arXiv preprint arXiv:2201.07520 (2022).

**Summary Of The Paper:**

This paper proposes to randomly mask past tokens in casual language models.


**Summary Of The Review:**

This paper presents a new casual masking objective, and carries out apple-to-apple comparisons with PaLM 128M, PaLM 1B. In comparison with this paper, [1] studied 6 different training objectives (including the most relevant casual masking method) and carried out comparisons with these six training objectives with 5 different scales: 125M, 355M, 1.3B, 2.7B, 6.7B. Due to seriously ignore related articles and casual masking method from [2] is pretty similar as this paper (approach of [2] might be better), I choose to reject this article.


*References*

[1]. Artetxe, Mikel, et al. "On the Role of Bidirectionality in Language Model Pre-Training." arXiv preprint arXiv:2205.11726 (2022).

[2]. Aghajanyan, Armen, et al. "Cm3: A causal masked multimodal model of the internet." arXiv preprint arXiv:2201.07520 (2022).

---

> ### Author Response · Authors · 2022-11-11
> **Author response to reviewer wa4A**
>
> We’d like to thank the reviewer for their helpful comments and feedback. We address the specific questions below.
>
>
>
> > Relation to CM3 and HYBUNI
>
>
> First, we explain how our method is largely orthogonal to CM3 and HYBUNI, and in fact, our method can be easily integrated into such methods:
> - CM3 and HYBUNI focus on introducing bidirectionality into causal masking objectives, where the masked tokens are moved to the end of the sequence in order to make it possible to attend to future positions from these masked tokens. The introduction of bidirectionality gives the model additional capabilities to attend to the future when predicting masked tokens.
> - On the contrary, our paper focuses exclusively on the setting of unidirectional language models. Unlike CM3 and HYBUNI, our approach focuses on improving autoregressive language models without altering the input sequence or changing the training objective. Instead, we only change the attention masks for unidirectional language models. In fact, we can incorporate our attention masking method into bidirectional models such as CM3 and HYBUNI, but it is beyond the scope of the paper and we leave it for future work.
>
> Additionally, to further address the reviewer's concerns, we have added new experimental results in Table 7 and Appendix A.4 of the updated paper:
>
> Table 7: We compare to a casual masking objective that uses masked tokens instead of attention masks. The results show that using attention masks is significantly better than using mask tokens in zero and few shot performance, which further validates the effectiveness of our approach.
>
> Appendix A.4: We also added comparison with a simpler$^*$ version of CM3/HYBUNI, where we split the sentence into [prefix, middle, suffix] and move [middle] to the end of the sequence. However, we did not see an improvement in zero- and few-shot learning (see Table 12). Note that another prior work [Bavarian et al., 2022] reported a similar finding that few-shot performance does not improve by moving the infill regions to the end of the context.
>
> $^*$ The implementations for CM3 and HYBUNI are not publicly available, so we were unable to exactly reproduce those methods. (Note: There are many hyperparameters in CM3 and HYBUNI: For example, the original method segments the sentence into multiple segments, and the number of segments itself is sampled from a Poisson distribution.)
>
>
> CM3 demonstrates impressive results on cross-modal generation and representation learning tasks. We believe combining CM3 and FCM could further improve the performance which we leave as an interesting future work.
>
>
> > Apple-to-apple comparisons with other models
>
>
> Previously, we weren’t able to compare to the large 8 billion parameters PaLM in an apple-to-apple setting due to limited compute resources, but we were able to complete those comparisons now. We have included additional results in Table 3 (finetuning) and Tables 4 and 5 (few-shot) of the updated paper.
> FCM 8B significantly outperforms 8B-model-size PaLM model's fine-tuning performance on all 8 SuperGLUE tasks, improving the score from 80.7 to 83.1. FCM 8B also improves the PaLM's SuperGLUE zero-shot performance from 61.6 to 64.0.
> Since our approach is built on top of PaLM, the comparison to PaLM is under the exact same settings with identical hyperparameters and training configurations. The results suggest that our approach is able to improve upon PaLM in a wide range of model sizes and tasks.
>
> Compared to other models such as UL2 and T5, the settings cannot be exactly the same due to differences in training objectives and model architectures. However, all these models are trained on the same dataset of C4, and when comparing to UL2 and T5, we always select the hyperparameter settings that are advantageous for the other models, such as a larger models (1.4B and 11B for T5, and 20B for UL2, vs. 1B and 8B for FCM) trained for longer epochs (1000B tokens for T5 and UL2, vs. 180B for FCM).
>
>
> We hope our response can address the concern of the reviewer, and we kindly ask the reviewer to participate in the discussion.
>
>
> [Bavarian et al., 2022] Efficient Training of Language Models to Fill in the Middle

---

> > ### Comment · Reviewer_wa4A · 2022-11-19
> > **Disagree with the author's announcement**
> >
> > I disagree with the author's announcement. The method proposed by this paper is extremely similar to the CM3 [2] and HYBUNI in [1]. In addition, this paper totally ignores these two related works.
> >
> > In their rebuttal, the author claims "CM3 and HYBUNI focus on introducing bidirectionality into causal masking objectives, where the masked tokens are moved to the end of the sequence." *This claim is wrong*. Please take a look at Table 1 in paper [1]. n_mask means how many tokens are masked, n_bidir means how many tokens are used for bidirectional prediction, n_prediction means how many tokens are put to the end of sentences for prediction.
> >
> > This HYBUNI method used n_prediction equals n (n is sequence length), n_bidirectional equals 0, and 15% of tokens have been masked. *n_prediction equals to total sequence length, which means nothing has been put to the end of sequence*. n_bidirectional equals to 0, which means nothing has been used for bidirectional prediction. *That is to say*, there are n tokens in a sequence, and 15% tokens have been masked and continue to use casual language modeling for prediction. *Is it extremely similar to this paper*?
> >
> >
> > [1]. On the Role of Bidirectionality in Language Model Pre-Training

---

> > > ### Author Response · Authors · 2022-11-19
> > > **Re: Disagree with the author's announcement**
> > >
> > > We respectfully disagree with the reviewer’s comment. In HYBUNI the number of  n-bidirectional tokens is 0, but the mask tokens are still used and moved to the end of the sequence, as stated in the paper [1]:
> > >
> > > > n_masks controls how many tokens are masked. Masked tokens are moved to the end along with their positional embeddings.
> > >
> > > HYBUNI [1] adopts a non-zero n_mask value and moves the masked tokens to the end of the sequence which allows the masked tokens to attend to the unmasked tokens in later positions. This operation introduces bidirectionality when predicting the masked tokens. As stated in the paper [1]:
> > >
> > > > Our framework allows us to vary the two notions of bidirectionality discussed above: n_bidir controls the weight of bidirectional attention, whereas n_mask and n_predict control the weight of bidirectional context.
> > >
> > > Our method, on the other hand, focuses on improving unidirectional language models without altering the order of the sequence and without introducing mask tokens. This makes our model complementary to HYBUNI.
> > >
> > > HYBUNI [1] categorizes language models and compares them, but FCM does not belong to any of these categories, we would like to emphasize two important differences of FCM that makes it work well in zero-shot, few-shot and fine-tuning.
> > > - **FCM does not alter the input sequence order** while HYBUNI alters the order.
> > > - **FCM does not use mask tokens** and instead employs attention masks to prevent tokens from being attended to.
> > >
> > > We note that **masked tokens can be applied in uni-directional LM without moving the masked tokens to the end of the sequence**.  We agree with the reviewer that it is important to compare our method to this baseline. To study the effect of using mask tokens, we compare to a casual masking objective that uses masked tokens instead of attention masks,  and present the results in Table 7. **The results show that using attention masks is significantly better than using mask tokens in zero and few shot performance, which further validates the effectiveness of our approach**.
> > >
> > >
> > > Per reviewer’s request, we also included a comparison with a simpler version of HYBUNI to study the effect of altering sequence, where we split the sentence into [prefix, middle, suffix] and move [middle] to the end of the sequence. However, we did not see an improvement in zero- and few-shot learning (see Table 12). **Similar to our observation, we note that according to Table 7 from [1], HYBUNI reduces the zero-shot performance while our method significantly improves zero-shot performance of uni-directional language models (see Figure 1, Table 2 and Figure 3)**. These results are consistent with other prior works [2] which also observed that altering sequence order by moving the infill regions to the end does not improve zero-shot and few-shot performance.
> > >
> > > We hope that our response addresses the concerns, but if not, please let us know.
> > >
> > >
> > >
> > > ### References
> > >
> > > [1] Artetxe, Mikel, et al. "On the Role of Bidirectionality in Language Model Pre-Training." arXiv preprint arXiv:2205.11726 (2022).
> > >
> > > [2] Bavarian, Mohammad, et al. "Efficient training of language models to fill in the middle." arXiv preprint arXiv:2207.14255 (2022).

---

> > > > ### Comment · Reviewer_wa4A · 2022-11-20
> > > > **Reply to the author**
> > > >
> > > > Thanks for the author's reply.
> > > >
> > > > (1). I know that HYBUNI method is slightly different from the method proposed by this paper. Author claims: "FCM does not alter the input sequence order while HYBUNI alters the order."
> > > > I would not say HYBUNI alter the input sequence order. It just moved masked tokens to end of sequence, which makes masked tokens attend to the backward information. I don't think this is similar to your experiments "we split the sentence into [prefix, middle, suffix] and move [middle] to the end of the sequence". Given this method is so similar to HYBUNI, they don't have a direct comparison instead of having an experiments of altering sequence order.
> > > >
> > > > (2). Since the method of this paper is very simple, it is hard to say that it is innovative from method perspective, so I expect this paper to have more detailed analysis. For example, T5 paper [2] explores different training data, different objectives, different masking techniques, varying the corruption rate, different corrupting spans, etc. As a comparison, this paper lacks detailed analysis. For example, Bigscience paper [1] found that causal decoder-only models trained on an autoregressive language modeling objective exhibit the strongest zero-shot generation. Models with non-causal visibility on their input trained with a masked language modeling objective performs best in finetuning. I guess HYBUNI may be better in finetuning. Given this method combine making with casual language modeling, could there be any new discoveries on above of big science explorations?
> > > >
> > > >
> > > > [1]. What Language Model Architecture and Pretraining Objective Work Best for Zero-Shot Generalization?
> > > > [2]. Exploring the Limits of Transfer Learning with a Unified Text-to-Text Transformer

---

> > > > > ### Author Response · Authors · 2022-11-20
> > > > > **Re: Reply to the author**
> > > > >
> > > > > We’d like to thank the reviewer for the timely response to our comments
> > > > >
> > > > > > Given this method is so similar to HYBUNI, they don't have a direct comparison instead of having an experiments of altering sequence order.
> > > > >
> > > > > Our paper focuses on improving the zero-shot and few-shot capabilities for causal language models. As shown in Table 7 of [1], **HYBUNI decreases the zero-shot performance compared to a standard uni-directional language model**, while our method significantly improves the zero-shot and few-shot performance (see Figure 1, Table 2, Figure 3). Moreover, we note the authors of [1] have not released the source code or pre-trained model for HYBUNI, making it difficult for us to create an implementation from scratch during the author response period.
> > > > >
> > > > > We hope to emphasize that while both HYBUNIN and FCM combine masking with uni-directional language models, there are several major differences which are summarized below.
> > > > >
> > > > > | Property |  HYBUNI  |  FCM  |
> > > > > | :------------------ | :--------------: | :--------------: |
> > > > > | Move masked tokens to the end of sequence for non-causal prediction | ✅| ❌|
> > > > > |  Use mask tokens | ✅| ❌|
> > > > > | Use attention mask | ❌| ✅|
> > > > > | **Improve zero-shot performance** | ❌| ✅|
> > > > >
> > > > >
> > > > >
> > > > >
> > > > > > Since the method of this paper is very simple, it is hard to say that it is innovative from method perspective.
> > > > >
> > > > > To the best of our knowledge, our paper is the first to use random attention masks to improve the zero-shot and few-shot performance of language models. Regarding the simplicity of our method, we view it as an advantage of our method rather than a disadvantage, since the simplicity makes it possible to integrate our method into a wide range of other works, which can potentially benefit a wide range of models. Due to its simplicity and ease of implementation, Reviewer 8z3 has commented: "I expect this to become a part of the pretraining toolbox for causal LLMs."
> > > > >
> > > > > >  Given this method combine making with casual language modeling, could there be any new discoveries on above of big science explorations?
> > > > >
> > > > > As the reviewer suggests, [2] discovers that “causal decoder-only models trained on an autoregressive language modeling objective exhibit the strongest zero-shot generalization.” **We note our method further improves the zero-shot and few-shot capabilities for causal decoder-only models, which is the central focus of our paper**. Furthermore, as FCM significantly improves fine-tuning performance (see Figure 1, Figure 3), applying FCM to the multi-task prompt fine-tuning setting in [2] should further improve zero-shot performance.
> > > > >
> > > > > We agree with the reviewer that studying other properties of language models, such as which objective is best for fine-tuning, is also interesting and valuable. However we believe that’s beyond the scope of this paper and we will leave it for future work.
> > > > >
> > > > > We hope that our response addresses the concerns, but if not, please let us know.
> > > > >
> > > > >
> > > > > ### References
> > > > >
> > > > > [1] Artetxe, Mikel, et al. "On the Role of Bidirectionality in Language Model Pre-Training." arXiv preprint arXiv:2205.11726 (2022).
> > > > >
> > > > > [2] Wang, Thomas, et al. "What Language Model Architecture and Pretraining Objective Work Best for Zero-Shot Generalization?." arXiv preprint arXiv:2204.05832 (2022).

---

> ### Author Response · Authors · 2022-11-16
> **Request for Discussion**
>
> Dear Reviewer wa4A,
>
> We'd like to thank the review again for the detailed review and constructive suggestions on our paper. We believe that we have addressed all of you concerns raised in the your review in our response and the revised paper. Could you please let us know if you have any additional concerns or questions? We would be happy to provide further revisions or experiments to address any remaining issues, and would really appreciate a response from you on the points that we raised before the end of the discussion period.

---

### Official Review · Reviewer_r8mR · 2022-10-25

**Confidence:** 3
**Correctness:** 3
**Technical Novelty And Significance:** 3
**Empirical Novelty And Significance:** 3
**Recommendation:** 6

**Clarity, Quality, Novelty And Reproducibility:**

It would be great if the authors could elaborate more on the disadvantage of randomly making out previous tokens in terms of the actual engineering implementation. It would be great if the authors could clarify if there were any misunderstanding.

Besides this part, the paper is well-written and easy to follow, with clear explanations of the technical parts. The experimental set-up is well explained, and the results are also well portrayed with convincing results, explanations, and ablations.



**Strength And Weaknesses:**

Strengths
1. The method is very simple and well explained in the method section. Experimental details are also well explained.
2. Different ablations of which factor of the proposed method contribute to the performance enhancement are provided.
3. Proposed method boosts performance in the zero-shot setting as well as when fine-tuned to perform the downstream task.

Weaknesses
1. While the proposed method does not add any computational cost in theory, since previous tokens are randomly masked out, won't there be an engineering overhead since the previous hidden states cannot be cached which would have remained unchanged if the previous token weren't masked out otherwise?

**Summary Of The Paper:**

This paper proposes a simple method of randomly masking past tokens during causal language modeling that boosts zero-shot capabilities  and fine-tuning results by a non-trivial margin. The problem motivation is capturing the best of both decoder and encoder LMs. While previous methods achieve this as well, the authors assert that the proposed method in this paper does not add any additional computational cost.

**Summary Of The Review:**

The authors propose a very simple methodology of simply masking out random tokens during causal language modeling which proves to be effective at enhancing both zero-shot capabilities and the downstream fine-tuning results. I highly recommend this paper be published at this conference. However, it would be great if the authors could elaborate more on the question of the proposed method in terms of the engineering implementation aspect.

---

> ### Author Response · Authors · 2022-11-11
> **Author response to reviewer r8mR**
>
> First of all we’d like to thank the reviewer for their constructive comments and positive view of our paper. We address the specific questions below.
>
>
> > Discuss engineering overhead
>
>
> FCM has very minimal engineering overhead to implement on top of standard causal Transformers. Due to its simplicity and ease of implementation, Reviewer 8z3 has commented: "I expect this to become a part of the pretraining toolbox for causal LLMs."
>
> More specifically, transformers rely on an attention mask matrix to control the attention, and causal Transformers apply a lower triangular attention mask matrix to ensure that each position only attends to previous locations. Our implementation simply masks out this attention matrix by setting some of its entries to zero during training time. To illustrate our implementation, we provide a NumPy implementation for computing the FCM attention matrix here.
>
> ```
> def get_fcm_attention_mask(batch_size, seq_len, min_fcm_ratio, max_fcm_ratio):
>     # Standard causal masking for autoregressive language model
>     # causal_mask[i, j, k] controls whether position j can attend to position k
>     # in the ith example of the batch
>     causal_mask = np.tril(np.ones((batch_size, seq_len, seq_len), dtype=bool))
>     # Sampling random FCM mask ratio independently for each example in batch
>     fcm_ratio = np.random.uniform(min_fcm_ratio, max_fcm_ratio, size=batch_size)
>     # Sampling FCM mask according to the sampled ratio
>     fcm_mask = np.random.uniform(size=(batch_size, 1, seq_len)) > fcm_ratio
>     # Disallow masking on BOS token
>     fcm_mask[:, :, 0] = True
>     # Combine standard causal mask with FCM
>     return np.logical_and(fcm_mask, causal_mask).astype(np.float32)
> ```
>
> During inference time, similar to many other language model regularization techniques, we do not apply any forgetful causal masking in order to maximally retain the information from the input. Hence, the inference implementation is exactly the same as that of a normal causal language model, and any caching operations for causal Transformers applies equally to FCM without any modifications.
>
> We hope our response can address the concern of the reviewer. In case there are any additional questions or concerns, please let us know by responding here.

---

> ### Author Response · Authors · 2022-11-16
> **Request for Discussion**
>
> Dear Reviewer r8mR,
>
> We'd like to thank the review again for the detailed review and positive view on our paper. We believe that we have addressed all of you concerns raised in the your review in our response and the revised paper. Could you please let us know if you have any additional concerns or questions? We would be happy to provide further revisions or experiments to address any remaining issues, and would really appreciate a response from you on the points that we raised before the end of the discussion period.

---

> > ### Comment · Reviewer_r8mR · 2022-11-16
> > **Response to Author Rebuttal**
> >
> > Thank you for the clarification regarding the concerns of additional computational costs of the proposed method. I'll leave my rating as it is since it reflects my best judgment of the paper.

---

### Official Review · Reviewer_8z3G · 2022-10-29

**Confidence:** 4
**Correctness:** 3
**Technical Novelty And Significance:** 2
**Empirical Novelty And Significance:** 3
**Recommendation:** 6

**Clarity, Quality, Novelty And Reproducibility:**

**Clarity**: The ideas are presented reasonably clearly and in a structured manner.

**Quality**: The quality of the paper is mostly good, with some minor issues pointed out below which are easily fixable.
- some grammatical mistakes and typos.
- Appropriate use of capitalization in the ‘References’ section.

**Novelty**:
- Low on technical novelty. The simple, novel, scalable pre-training masking tactic seems to have a broad minor to moderate impact on a variety of tasks. I expect this to become a part of the pretraining toolbox for causal LLMs.

**Reproducibility**:
- Should be reproducible (subject to availability of computational resources).


**Strength And Weaknesses:**

## Strengths
- Simple and scalable approach: no extra parameters or computations are added to pre-training using causal LM objectives.
- Extensive experimentation
- Performance improvements (modest) on a wide variety of tasks

## Weaknesses

### Training
- __(W.0)__ Kindly share the computational resources used, memory requirements and the training time.

### Results and Analysis
- __(W.1)__ In Table 1, along with the mean of the 3 random realizations, the spread should be provided as well, given that many of the performance numbers are within a single percentage point.
- __(W.2)__: The performance of FCM appears to be better than PaLM, but not significantly and consistently so. For significance, I’m using a criterion of at least 1% in absolute terms. It will be good if the authors can shed some light on the following:
- __(W.2.a)__ __0-shot__: In 8/19 tasks, the performance improvement is < 1%.
- __(W.2.b)__ __1-shot__: In 9/ 19 tasks, the performance improvement is < 1% and 2/19 times it’s worse.
- __(W.2.c)__ __few-shot__: In 4/ 19 tasks, the performance improvement is < 1% and 4/19 times it’s worse.
- __(W.2.d)__ Scaling trends: If we look at the consistency of performance improvement as the number of shots increase, there are very few tasks, on which FCM consistently stays better than PaLM by at least a percentage point. On the NLU tasks, FCM seems to lose its advantage as the number of shots increase, sometimes becoming much worse.
- __(W.3)__ __Classification finetuning__: results on SuperGLUE dev set only show significant improvement on 4 out of 8 tasks, with majority of the gains observed on WSC and CB. The authors should discuss the variability in performance gains.
- __(W.4)__ __Scalability, 0-shot__: In Table 4, FCM 1B results seem better than FCM 8B on CB, RTE, and WiC tasks. Kindly discuss.
- __(W.5)__ __Scalability, k-shot__: In Table 5, the performance of both PaLM and FCM sometimes, and more often for FCM, degrades as the number of shots increases. Kindly discuss.
- __(W.6)__ __Ablation of mask ratio__: Irrespective of the average across all the tasks, FCM[0,0.15] is better only on 2 out of 8 tasks in the 0-shot setting and 3 of 8 tasks in the 1-shot setting. This shows that the choice of mask ratio is task dependent. Kindly discuss.
- __(W.7)__ __Comparison with Dropout__: Contrary to the claim of dropout hurting FCM performance, it improves in 3 of 8 tasks. It improves performance by 10.2 absolute % points for the MultiRC task. It seems that the benefit of dropout is task dependent.


**Summary Of The Paper:**

The paper proposes a novel, simple, scalable and efficient pre-training approach for causal, decoder-only language models. The proposed approach is called Forgetful Causal Masking (FCM) and it randomly masks out past tokens. FCM pre-training is applied to PaLM to demonstrate significant improvement in the 0-shot performance on the SuperGLUE benchmark. FCM is also tested on a variety of 0-shot and few-shot tasks - PIQA, ARC, OpenBookQA, Winograd, WinoGrande, ANLI, StoryCloze, and LAMBADA tasks to demonstrate consistent performance improvement across such diverse tasks.


**Summary Of The Review:**

The paper proposes a simple and scalable pre-training tactic - of causal random masking for causal language modeling. It doesn’t require additional parameters or computations. Extensive experimental evaluation is performed. While the performance gains seem modest to moderate, and somewhat inconsistent, they are largely positive across a large range of tasks. Though the technical novelty is limited, I expect the proposed mechanism to become a part of the pretraining toolbox for causal LLMs.

---

> ### Author Response · Authors · 2022-11-11
> **Author response to reviewer 8z3G**
>
> First of all we’d like to thank the reviewer for their detailed and constructive comments, which helped improve the clarity of our paper. Moreover, we are happy to see the reviewer's positive view of our paper. We address the specific questions below.
>
>
> > Share the computational resources used, memory requirements and the training time.
>
>
> We have updated the paper to include this information in Appendix A.1. The experiments are conducted on TPU v4 accelerators, where each TPU v4 has 32 GiB memory available. For 128M models, training 180B tokens takes 18 hours on 64 accelerators. For 1B models, training on 180B tokens takes 25 hours on 256 accelerators.  The training of 8B models on 180B tokens takes 100 hours on 512 accelerators.
>
>
> > Include the spread of results on three random realizations.
>
>
> We thank the reviewer for the suggestion and we have included the evaluation results on three random realizations in Appendix A.3.
>
>
> > Performance of FCM appears to be better than PaLM, but not significantly and consistently so on all 19 tasks.
>
>
> We would like to emphasize that FCM performs significantly better than PaLM on the SuperGLUE benchmark, both in finetuning (Table 3) and few-shot learning (Table 4 and Table 5), where the absolute improvement is by 2.4%, 2.4%, and 2.8% on zero-, one-, and five-shot and 2.5% on finetuning with 8B model and similarly the absolute improvement is by 3.5%, 2.6%, 2.7%, and 1.7% with 1B model. We acknowledge that there are some tasks where our model does not significantly outperform PaLM, but we believe the comprehensive comparison and the overall improvement on a majority of the tasks show the effectiveness of our approach.
>
>
> > Discuss why SuperGLUE finetuning shows significant improvement on 4 out of 8 tasks.
>
>
> Table 3 provides finetuning performance of  larger 8B models.  FCM 1B outperforms PaLM 1B significantly on 4 out of 8 SuperGLUE tasks, and FCM 8B significantly outperforms PaLM 8B on 8 out of 8 SuperGLUE tasks, improving the score from 80.7 to 83.1. This shows that FCM improves PaLM on representation / finetuning and scales well to large models. On the 1B model, the majority of the gain comes from CB, RECORD, RTE and WSC, while on the 8B model, there is a significant improvement on all eight tasks. We agree with the reviewer that the improvements on specific tasks depend on the exact characterizations of the tasks, which aligns with findings from prior work including PaLM and GPT-3.
>
>
> > Discuss why FCM 1B results seem better than FCM 8B on three tasks (Table 4)
>
>
> Since the average results across eight SuperGLUE few-shot tasks and fine-tuning tasks of FCM 8B are significantly better than FCM 1B, we believe that FCM 1B outperforms FCM 8B on three few-shot tasks due to exact characterizations of the tasks.
> Similar observations can be found in the PaLM and GPT-3, e.g. PaLM 64B outperforms 540B on Copa and CB. We believe future studies of this can further improve large language models.
>
>
> > Discuss why the performance of PaLM and FCM can degrade with the number of shots increasing
>
>
> While we are not able to characterize the reason for this phenomenon exactly, we hypothesize that the language model could be sensitive to the prompt setup of certain tasks, and therefore in some cases adding more shots could result in decreased performance. We note that this phenomenon is also observed in prior works such as GPT-3 and PaLM. The exact characterization of this phenomenon is beyond the scope of this paper and we will leave it to future works.
>
> However, we hope to emphasize that the average performance over many tasks increases reliably with the number of shots for both the 1B model and 8B model, as seen in Table 2 of the paper. With the updated results of few shot performance of the 8B model, we see that the performance improvement of FCM over PaLM is significant on the majority of tasks.
>
>
> > Discuss whether the optimal mask ratio of FCM could be task-dependent
>
>
> We believe that our chosen mask ratio is not optimal due to limited hyperparameter tuning. Subject to the compute resources available, we evaluated different mask ratio choices in few-shot and finetuning experiments. We use [0, 15%] since on average it performs better than other mask ratios across SuperGLUE few-shot and SuperGLUE finetuning. Further tuning of the mask ratios based on a wide range of tasks could further improve the results.

---

> > ### Author Response · Authors · 2022-11-11
> > **Author response to reviewer 8z3G (continue)**
> >
> > > Discuss whether the benefit of dropout is task-dependent
> >
> >
> > Thank you for pointing out the interesting result that, while using dropout alone is harmful to training large language models, using FCM and dropout together improves results on some tasks
> > The results in Table 7 show that using dropout during large language models pre-training is harmful, decreasing the score from 55.7 to 55.4, which aligns with findings from prior work. In contrast, combining dropout with FCM can improve the performance of PaLM from 55.7 to 57.7 on SuperGLUE average score, indicating that FCM and dropout can be complementary. We leave further studies of this as interesting future work.
> >
> > Note that on average, we found that using only FCM performs slightly better than combining dropout+FCM, showing the effectiveness of FCM on the next-token prediction task with randomly selected past tokens masked out. Since FCM and dropout are practically orthogonal, we leave it to future work to optimally combine FCM and dropout together.
> >
> >
> > We thank the reviewer for bringing up these detailed discussion points, and we have included the discussion points in the revision.  We hope that our response addresses all of your questions, but if not, please let us know.

---

> ### Author Response · Authors · 2022-11-16
> **Request for Discussion**
>
> Dear Reviewer 8z3G
>
> We'd like to thank the review again for the detailed review and positive view on our paper. We believe that we have addressed all of you concerns raised in the your review in our response and the revised paper. Could you please let us know if you have any additional concerns or questions? We would be happy to provide further revisions or experiments to address any remaining issues, and would really appreciate a response from you on the points that we raised before the end of the discussion period.

---

> > ### Comment · Reviewer_8z3G · 2022-11-28
> > **Acknowledgment**
> >
> > Dear Authors:
> >
> > Thanks a lot for your detailed response. I've factored this into my review. I have no further questions. All the best with the paper decision (and your research in general).
> >
> > Regards

---

### Official Review · Reviewer_pAfF · 2022-11-04

**Confidence:** 5
**Correctness:** 4
**Technical Novelty And Significance:** 2
**Empirical Novelty And Significance:** 2
**Recommendation:** 3

**Clarity, Quality, Novelty And Reproducibility:**

This clarity of this paper is clear, and the method mentioned in this paper is easy to reproduce.  The novelty of this paper is limited.

**Strength And Weaknesses:**

Strength:
1. The proposed method is simple and easy to reproduce.
2. Experimental results show the proposed method can effectively improve the LLMs in both zero- and few-shot tasks.

Weaknesses:
1. This method can be seen as a variation of schedule sampling (Bengio et al., 2015) or a simplified version of UniLM (Dong et al., 2019). The novelty of this paper is limited.
2. On the other hand,  the authors should give more analysis about why this method can work in LLMs. The training set of pre-training models contains a lot of noise data. I wonder if the proposed method is necessary.


Bengio et al., 2015. Scheduled Sampling for Sequence Prediction with Recurrent Neural Networks.
Dong et al., 2019. Unified language model pre-training for natural language understanding and generation.

**Summary Of The Paper:**

In this paper, the authors proposed a technique to improve zero-shot and few-shot capabilities for Large language models (LLM). Specifically, when performing the next token prediction task, the proposed method will randomly select past tokens masked out to fine-tune the LLMs. Experimental results show that the proposed FCM improves PaLM in both zero and few-shot tasks.

**Summary Of The Review:**

The novelty of this paper is limited.  Methods similar to this paper have been widely used in previous work. On the other hand,  the authors should give more analysis about why this method can work in LLMs.

---

> ### Author Response · Authors · 2022-11-11
> **Author response to reviewer pAfF**
>
> We’d like to thank the reviewer for their constructive comments and feedback. We address the specific questions below.
>
>
> > Relation to scheduled sampling
>
>
> Thanks for pointing out an interesting connection between our method and scheduled sampling [Bengio et al., 2015].
> We believe that our method is significantly different from scheduled sampling. Scheduled sampling addresses the gap between training and inference for autoregressive models, where during training, the model is always conditioned on the ground truth, but at inference time, the model is conditioned on its own generations. To address this, the scheduled sampling method randomly chooses to condition on model-generated tokens during training time.
>
> On the other hand, our approach uses attention masks to regularize and improve autoregressive models. From the perspective of train-inference gap, in contrast to scheduled sampling, our approach actually increases the gap by introducing extra attention masks during training time but not during inference time. From a practical perspective, FCM is orthogonal to (and can be easily combined with) scheduled sampling training, since the change in attention masking is agnostic of the tokens inputted into the model.
>
> Moreover, a recent work [1] shows that extending scheduled sampling to Transformers requires running the decoder in two passes during training, effectively doubling the amount of computation. On the contrary, our method does not change the computation time of standard Transformer models.
>
>
> > Relation to UniLM
>
>
> Our method is also largely orthogonal to UniLM. UniLM combines the masked token prediction objective with three types of attention masks: unidirectional, bidirectional, and prefix-LM. For bidirectional attention masks, UniLM also incorporates the next-sentence-prediction objective from BERT. Different from UniLM, our approach focuses only on the setting of unidirectional language models. Instead of altering the input sequence by replacing some tokens with the mask token (as in UniLM), we apply random attention masks to disallow future tokens from attending to some past tokens. Our approach improves unidirectional language models agnostic of its input and training objective, so it can be easily combined with the unidirectional LM objective of UniLM.
>
> Additional Experiment (Table 7): Despite the orthogonality of our method and UniLM, the unidirectional masked token prediction part from UniLM can be considered a different approach to achieve the same goal of regularizing and improving unidirectional language models. Therefore, we have conducted additional experiments on unidirectional language models where we use mask tokens to replace part of the input instead of applying attention masks as in our approach (see Table 7 in the updated paper). The new results show that using attention masks is significantly better than using mask tokens in zero- and few-shot performance, which further validates the advantage of our approach.
>
> Moreover, prior work [3] discovers that UL2 outperforms UniLM in zero-shot and few-shot evaluation settings. In this paper, we compare our approach to UL2 in Table 4, which is trained on the same dataset with a larger model. Despite using a much smaller model, our approach can still outperform UL2 in few-shot and zero-shot tasks.
>
>
>
> > More analysis about why this method can work in LLMs.
>
>
> Our intuition of why FCM works well is that, by randomly masking out past tokens from attention, we can force the model to pay more attention to a broader range of the context, thereby preventing the model from relying on shortcuts such as frequently co-occurring token pairs.
>
>
> > The training set of pre-training models contains a lot of noise data. I wonder if the proposed method is necessary.
>
>
> The C4 dataset used to pre-train our model is a widely-used open-source dataset used to train language models [1, 2, 3]. It is well-filtered to contain reasonably clean and natural English text [1]. Hence we believe that the evaluation of our approach on this dataset is reliable. Given our limited computational resources, we leave it to future work in scaling our method up to even larger datasets [4].
>
>
> We hope our response can address the reviewer's concerns, and we kindly ask the reviewer to participate in the discussion.
>
>
> References
>
> [1] Raffel, Colin, et al. "Exploring the limits of transfer learning with a unified text-to-text transformer." J. Mach. Learn. Res. 21.140 (2020): 1-67.
>
> [2] So, David, et al. "Searching for Efficient Transformers for Language Modeling." Advances in Neural Information Processing Systems 34 (2021): 6010-6022.
>
> [3] Tay, Yi, et al. "Unifying Language Learning Paradigms." arXiv preprint arXiv:2205.05131 (2022).
>
> [4] Gao, Leo, et al. "The pile: An 800gb dataset of diverse text for language modeling." arXiv preprint arXiv:2101.00027 (2020).

---

> ### Author Response · Authors · 2022-11-16
> **Request for Discussion**
>
> Dear Reviewer pAfF
>
> We believe that we have addressed all of you concerns raised in the your review in our response and the revised paper. Could you please let us know if you have any additional concerns or questions? We would be happy to provide further revisions or experiments to address any remaining issues, and would really appreciate a response from you on the points that we raised before the end of the discussion period.

---

> ### Author Response · Authors · 2022-11-23
> **Follow up Request for Discussion**
>
> Dear Reviewer pAfF,
>
> We believe we have addressed the concerns you raised in your review, and we are eager to hear your feedback. Could you please let us know if you have any additional concerns or questions? We are looking forward to your engagement in the discussion.

---

### Author Response · Authors · 2022-11-11
**General response**

First of all, we’d like to thank the reviewers for their constructive comments and feedback. We thank the reviewers for appreciating our results and method, and thinking that “the proposed method is simple” and “can effectively improve the LLMs”(reviewer pAfF), “I expect this to become a part of the pretraining toolbox for causal LLMs” (reviewer 8z3G), “very simple and well explained” and “boost the performance of LLM” (reviewer r8mR), and “well written and easy to understand” (reviewer wa4A).

We have conducted additional experiments and ablation studies to address the reviewers' concerns. We include these results in the updated version of the paper, where the changes are highlighted in red.

We include an additional baseline of a 8B PaLM model that is trained on the C4 dataset in Table 3 (finetuning) and Table 4 and 5 (few-shot) . This extra result gives us an apple-to-apple comparison of our approach and PaLM on the 8 billion parameter model. We can see that FCM 8B outperforms PaLM 8B significantly both in the zero- and few-shot evaluation setting and the fine-tuning setting.

Moreover, as suggested by reviewer pAfF and wa4A, we include another baseline comparison where we use masked tokens instead of attention masks for the causal language modeling objective. This can be viewed as a special case of UniLM [Dong et al., 2019], or a variant of the causal masking objective in CM3 [Aghajanyan et al., 2022] without altering the sequence order to introduce bidirectionally. We present the results in Table 7. We can see that our approach outperforms the mask token approach in all of 0-shot, few-shot and fine tuning settings, further validating the improvement of our method.

We hope our response can address the concern of the reviewers, and we’d like to kindly ask the reviewers to participate in the discussion.

[Dong et al., 2019]. "Unified Language Model Pre-training for Natural Language Understanding and Generation", NeurIPS 2022

[Aghajanyan et al., 2022]. "CM3: A Causal Masked Multimodal Model of the Internet." arXiv preprint arXiv:2201.07520 (2022).

---

### Decision · Program_Chairs · 2023-01-20

**Decision:**

Reject

**Justification For Why Not Higher Score:**

This is a hard decision. The current submission reads more like a methodology paper with solid performance gain. The reviewers hope to see more in-depth discussion about the insight and conjecture why the proposed method works, and theory/reasoning/experiments verify the conjecture. Such discussion will entertain more ICLR participants and uplevel the impact of this submission.

**Justification For Why Not Lower Score:**

N/A

**Metareview: Summary, Strengths And Weaknesses:**

This submission proposed a simple yet effective pre-training approach, Forgetful Causal Masking (FCM), for causal, decoder-only Large language models (LLM). FCM randomly masks out past tokens during causal language modeling in next token prediction task. Experimental results show that the proposed FCM improves PaLM consistently in both zero and few-shot tasks on diverse benchmarks including SuperGLUE, PIQA, ARC, OpenBookQA, Winograd, WinoGrande, ANLI, StoryCloze, and LAMBADA.  It is popular area trying to capture the best of both decoder and encoder LMs. The authors demonstrate that the proposed method does not add any additional computational cost.

Strength of the submission:
1. The proposed method is simple and easy to reproduce. No extra parameters or computations are added to LLM, which allows FCM to be applied to broader use cases.
2. Experimental results also show the proposed method effectively improve the LLMs in both zero- and few-shot tasks. Experiments are done on a variety of popular benchmarks in zero shot and few shot tasks. The consistent improvement and simple techniques make it almost no doubt the benefit of FCM on causal LLM is generalizable.
3. Several ablations to explore which which factor of FCM contributes to the performance gain.

Weakness/missing of the submission:
1. Several reviewers raise the concern about technical novelty as the method is relatively simple. Novelty is a relatively subjective metric, and simple but effective method is still valuable. However, in a highly competitive venue such as ICLR, we do hope the accepted paper can enlight the community with future research direction and demystify why the method works or not works, beyond showing promising performance results. The submission proposes a simple method that improves performance consistently across diverse benchmarks. It is convincing that the proposed method is effective. However, the reviewers prefer to see more discussion about what makes the simple method work (not only ablation showing how each component contributes to the final results, but also what's the intution behind the simple design, what's the conjecture why the approach is effective, any patterns the authors observed in the experiments that can/can't be applied to other tasks or type of LLMs, any theory or experiments that can support the conjecture, etc.). With such discussion, the impact of the submission will be signifcantly upleveled.

The authors and several reviewers have disagreement about if FCM is similar to existing methods such as CM3/HYBUNI. The area chair believes that the difference between proposed method and existing ones is often subjective but less decisive. It's more important to provide empirical evidence supporting efficacy of the overall method, insight and conjecture why the method works, and theory/reasoning/experiments verify the conjecture.